# Airborne Millimeter-Wave InSAR Terrain Mapping Experiments Based on Automatic Extraction and Interferometric Calibration of Tie-Points

Bin Zhang *, Futai Xie , Liuliu Wang, Shuang Li, Lideng Wei and Liang Feng

Beijing Institute of Radio Measurement, Beijing 100854, China
* Correspondence: zbhian123@htbirm.cn

**Abstract:** Limited by meteorological conditions, airspace, complex terrain and other factors, airborne millimeter-wave InSAR will inevitably face the situation of no control point layout when acquiring terrain data in the difficult mapping areas in Southwest China, which increases the difficulty of subsequent data processing. Moreover, the layout of control points in difficult mapping areas consumes a lot of manpower and time, which is not suitable for large-scale high-precision topographic mapping. To solve these problems, this paper proposes an automatic extraction of tie-points and interferometric calibration technology based on tie-points. This technology develops the automatic extraction algorithm of tie-points based on SAR + SIFT + RANSAC to obtain evenly distributed tie-points of adjacent images, and uses the evenly distributed tie-points as real known points to recalibrate the interference parameters, then carries out elevation transfer and elevation inversion through the tie-points of overlapping areas, thus realizing high-precision mapping without control points for airborne millimeter-wave InSAR. This paper uses measured data to verify the technology, and compares it with the areas with control points and marking points. The comparison results of elevation accuracy prove the feasibility and effectiveness of this method. This paper also discusses the difficulties in the treatment of typical areas, such as water areas, urban areas and mountain areas, and gives reasonable solutions that have good engineering application value.

**Keywords:** interferometric synthetic aperture radar (InSAR); terrain mapping; tie-points; automatic extraction; interferometric calibration; mapping without ground control points

## 1. Introduction

Interferometric synthetic aperture radar technology (InSAR) makes use of phase differences between different perspectives of SAR complex images and the interference imaging geometry relationship to obtain three-dimensional elevation information [1,2]. InSAR technology has many advantages, such as high resolution, all-day, all-weather remote sensing imaging ability, high efficiency, high accuracy, large width of surveying and mapping, low cost and so on, and it has been recognized as a powerful way to obtain topography data for a large bandwidth of surveying and mapping and three-dimensional information under all weather conditions [3,4]. In addition, InSAR technology is widely used in moving target detection, surface change detection, glacier motion detection, natural disaster forecasting (such as earthquake, volcano and landslide monitoring), land resource classification, military reconnaissance and other fields [5–7].

Compared with spaceborne InSAR, airborne InSAR has the advantages of strong flexibility, high revisit frequency and easy real-time processing [8,9]. Meanwhile, compared with low-band, the millimeter-wave interferometric InSAR system has higher resolution imaging, shorter baseline with rigid structure, smaller volume, lighter weighting, easier installation and images that contain more details of targets' characteristics. In recent years, with the continuous development and improvement of the technology of airborne millimeter-wave InSAR, it has gradually become a widely used mapping method [10,11].

Likewise, airborne millimeter-wave interferometric synthetic aperture radar (InSAR) has the capability of high-precision geographic mapping, and it needs to deploy a certain number of corner reflectors as control points to achieve terrain height inversion in large-scale geographic mapping. Although the layout of corner reflectors can ensure the elevation accuracy to a satisfactory degree, it is difficult and costly to deploy them [12,13]. On the one hand, the layout of corner reflectors consumes a lot of manpower and material resources. Due to the inconvenience of transportation, it is more difficult to deploy corner reflectors in high mountains, hills and other complex terrain areas [14–16]. On the other hand, due to the complexity of climate and flight airspace, the layout of corner reflectors is often conflicts with the flight of radar carrier aircraft, which is difficult to coordinate and consumes a lot of time. In addition, corner reflectors cannot be safely deployed in a real combat environment, which greatly limits the application of airborne InSAR terrain mapping technology in the military field. Therefore, it is particularly important to study the mapping processing technology of large uncontrolled areas [17,18].

At present, researchers have carried out a lot of research on airborne InSAR topographic mapping technology with few or no ground control points and plenty of tie-points. First of all, in the study of tie-point selection, the SIFT algorithm was widely used in the registration of optical images [19,20], but this method was inefficient in the registration of SAR images because of the inherent speckle noise in SAR images. Due to the characteristics of SAR images, the SAR-SIFT algorithm optimized the tie-point selection process and had higher performance on tie-point selection and SAR image registration, but the selected points were more focused on typical artificial targets [21]. Additionally, the OS-SIFT algorithm was developed to register the SAR images and optical images, but it had little application in the InSAR mapping domain [22]. In [23,24], the selection of tie-points was based on the self-similarity criterion and the coherence of the master and slave images, and then the tie-points were matched and eliminated based on the RANSAC algorithm, which can select a certain number of points, but the algorithm has not been further applied in large-area and large-scale InSAR mapping. Moreover, the algorithms mentioned above have a relatively obvious problem: that the selected tie-points are more concentrated on the artificial target, which is not evenly distributed in the image. When the tie-points are used in InSAR mapping, the even distribution of tie-points is necessary to carry out error avoidance and to obtain high-precision InSAR 3D measurement results. Because pixel position errors and elevation errors are unavoidable in tie-points, the height error of InSAR will be expanded when height transfer and calibration is carried out. Additionally, the quality of tie-points (InSAR height accuracy, coherence) is an order of magnitude lower than that of the control points. The tie-points we select should be distributed on the artificial target as little as possible, at the same time, they should be evenly distributed in relatively flat places to guarantee the quality of the tie-points. Based on the above analysis, on the basis of the existing algorithm, it is necessary to make a suitable point selection strategy to ensure that the points are evenly distributed and as flat as possible in the overlapping areas. There is also a problem in that tie-points cannot be automatically extracted in batches [25,26]. At this time, blocking in overlapping areas is a good strategy, which ensure the selection of even tie-points and decreases the number of tie-points on the artificial features.

In addition, researchers have developed a variety of InSAR processing algorithms based on tie-points [27–29]. These methods mainly adopt the SIFT algorithm to extract the tie-points, as this algorithm is more suitable for optical images and has implicit efficiency of registration among SAR images, and the block adjustment mode in the algorithm is still needs to be verified in high-precision and large-scale InSAR geographic mapping of large areas. Huang et al. [30] proposed an InSAR data processing algorithm based on block adjustment., This method used few ground control points and plenty of tie-points to carry out the inversion of InSAR height, but it was not effectively verified in large-scale topographic mapping. Wang et al. [31] proposed an interferometric calibration algorithm based on a single corner reflector in two converse flights, but it has poor practicability. Li et al. [32] proposed an airborne millimeter-wave InSAR data processing flow based

on a backward projection algorithm, and used few ground points to calibrate the InSAR system error and carry out the height inversion process, which still needs to be further verified in the processing of SAR data from different sorties. Nevertheless, these methods combine ground control points and tie-points to run the processing. Some flight sorties have no ground points because of uncertainty about flight plans, bad weather, human costs, etc., and have an influence on the InSAR 3D product used with the above methods. At present, a calibration strategy only based on tie-points is still lacking. Therefore, it is necessary to study the high-precision terrain mapping processing technology of airborne millimeter-wave InSAR based on the automatic extraction and calibration of tie-points.

Based on the above analysis, this paper proposes a mapping technology without control points for airborne millimeter-wave InSAR based on block adjustment. Firstly, the tie-points, based on the SAR-SIFT-RANSAC algorithm, are automatically and evenly extracted in the overlapping area between adjacent data blocks within and between the flight sorties by image blocking and filtering of tie-points. Then, using the well-calibrated interferometric parameters from the calibration areas, we adopt the elevation data blocks to obtain the unknown elevation of adjacent data blocks with tie-points. Furthermore, the tie-points are used as control points to calibrate the interferometric parameters, thus completing the elevation inversion of adjacent data blocks within and between strips in different flight sorties, and effectively obtaining the airborne millimeter-wave InSAR mapping results without ground control points.

The rest of this paper is organized as follows. Section 2 introduces the basic principle of airborne InSAR, analyzes current problems in InSAR mapping and introduces the automatic extraction technology of tie-points and interferometric calibration based on tie-points throughout the whole InSAR procedure. Section 3 first introduces the tie-point extraction results within and between strips, then shows the effectiveness of the interferometric calibration of tie-points within and between flight sorties. In Section 4, we discuss the elevation analysis results of the proposed method, list three examples in typical areas (water bodies, urban areas and shadow areas) that affect the processing results and show their resolution to improve the efficiency of the proposal. Finally, conclusions, flight advice and future work of the paper are described.

## 2. Methods

### 2.1. Fundamentals of InSAR and Problem Description

In the ideal airframe coordinate system [8], the geometric model of InSAR processing is shown in Figure 1. Assuming that the position of antenna 1 is $A_1 = [x_a, y_a, z_a]^T$, and the position of antenna 2 is $A_2 = [x_b, y_b, z_b]^T$, then the interferometric baseline length is $B = \sqrt{b_x^2 + b_y^2 + b_z^2}$, where $b_x = x_b - x_a, b_y = y_b - y_a, b_z = z_b - z_a$. When taking antenna 1 as reference, that is, when $A_1 = [0, 0, 0]^T$, $b_x = x_b$, $b_y = y_b$ and $b_z = z_b$.

From Figure 1, the target position $T$ is:

$$T = [R\cos(\theta_{sq}), \sqrt{R^2 - \sin^2(\theta_{sq}) - (H - h)^2}, H - h]^T \tag{1}$$

The distance from antenna 2 to the target position is:

$$\left|A_2 T\right|_2 = R^2 + B^2 - 2b_x R\cos(\theta_{sq}) - 2b_y \sqrt{(R^2 \sin(\theta_{sq}) - (H - h)^2)} - 2b_z(H - h) \tag{2}$$

where $\theta_{sq}$ is the squint angle of antenna 1.

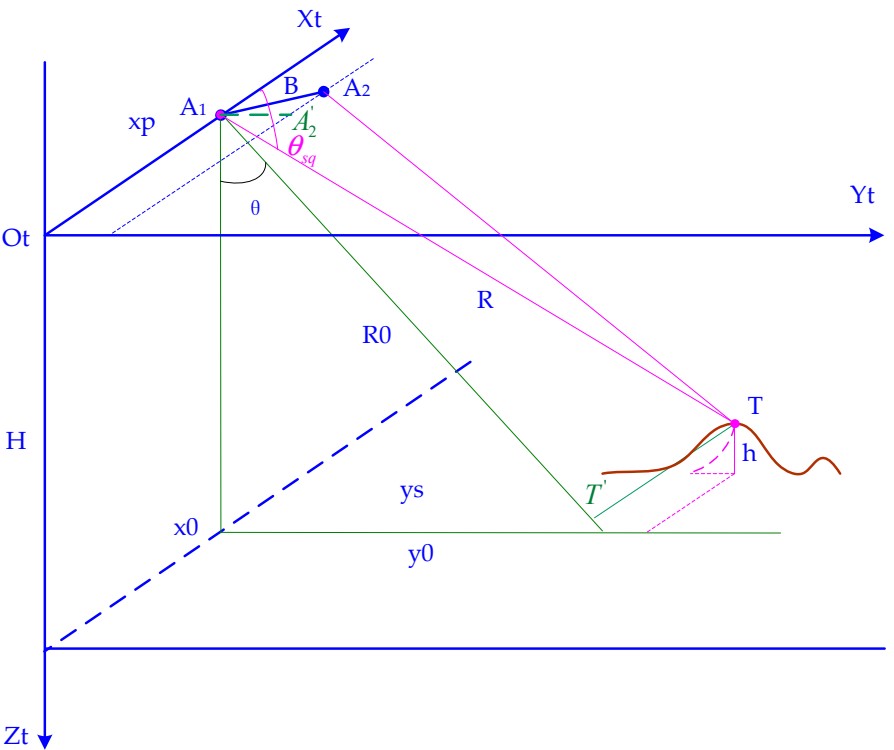

**Figure 1.** InSAR geometric model in ideal airframe coordinate system.

Given that the distance between antenna 1 and the target is $R$, the difference between the distance of antenna 1 and antenna 2 to the target is:

$$\Delta R \approx b_x \cos(\theta_{sq}) + b_y \sqrt{\sin^2(\theta_{sq}) - \cos^2(\theta)} + b_z \cos(\theta) \tag{3}$$

Then, we have:

$$h = H - R \left( \frac{-b_z [b_x \cos(\theta_{sq}) - \Delta R]}{b_y^2 + b_z^2} \pm \frac{b_y \sqrt{(b_y^2 + b_z^2)\sin^2(\theta_{sq}) - [b_x \cos(\theta_{sq}) - \Delta R]^2}}{b_y^2 + b_z^2} \right) \tag{4}$$

$$\Delta\phi = -\frac{2\pi}{\lambda}\Delta R \tag{5}$$

$$B_\perp = \sqrt{b_y^2 + b_z^2}, b_z = B_\perp \sin(\alpha), b_y = B_\perp \cos(\alpha) \tag{6}$$

The positive and negative signs in Equation (4) are determined according to the side-looking direction of the radar; the right side-looking is positive, and the left side-looking is the opposite.

According to the InSAR geometric model, the main parameters affecting the target elevation h are: vertical baseline length $B_\perp$, vertical baseline inclination $\alpha$, along track baseline $B_x$, interferometric phase $\Delta\phi$ and slant distance $R$.

When the radar carrier flies over the calibration field, the ground control points can be used for interferometric calibration to obtain the above error parameters that affect the elevation accuracy. However, these error parameters are not applicable to all flight sorties. Influenced by factors such as temperature, humidity and radar power on/off time, the above error parameters need to be recorrected in different flight sorties [17,33]. Otherwise, large elevation errors will be introduced, which will affect the large-area high-precision geographic mapping of airborne InSAR. The conventional method is to deploy the ground control points to correct these error parameters [10,12]. However, due to the influence of weather, terrain, operating environment and other factors in multiple sorties,

the deployment of control points requires a lot of manpower and material resources, and in some complex terrain, the deployment of control points cannot be completed. The key to solve this problem is the tie-points. In the experimental area without ground control points, especially for different sorties, the tie-points in the overlapping area of adjacent strip data blocks can be automatically extracted and used to correct the error parameters, and the elevation of unknown points can be calculated from the elevation of known points, so as to achieve the high-precision elevation inversion of large areas of airborne InSAR.

### 2.2. Automatic Extraction of Tie-Points

The calibration parameters obtained from the calibration field can be used for subsequent elevation inversion and other processing of the strips in the calibration field, while for other strips, the selected tie-points need to be used for elevation transfer to achieve the elevation inversion of the non-calibration filed.

The automatic extraction of tie-points includes the extraction of tie-points of adjacent data blocks within and between strips. The overlapping areas of adjacent data blocks in the same strip are essentially from the same original echo data, only the processing parameters of the data blocks are slightly different. Therefore, the amplitude and texture information of overlapping areas in the same strip are small, and the automatic extraction of tie-points is relatively easy. However, the overlapping area of adjacent data blocks between strips will produce geometric deformation due to the difference in SAR perspective, and the texture information and amplitude of the same object may vary greatly, which makes it difficult to automatically extract tie-points. To solve these problems, this paper develops an algorithm based on SAR + SIFT + RANSAC to automatically extract the tie-points of SAR images. The process of automatic extraction of tie-points of adjacent data blocks within/between strips is shown in Figure 2.

#### 2.2.1. Calculation of Overlapping Area

SAR images are generally large in airborne millimeter-wave SAR, and a single SAR image has at least $8192 \times 8192$ pixels, but the overlapping area between adjacent data blocks is about 15–30%. Therefore, it is unnecessary to extract feature points and search for tie-points on the whole SAR image. Calculating the overlapping area of two SAR images can effectively narrow the extraction range and improve the algorithm performance before the extraction of tie-points.

According to the longitude and latitude of the four corners of the two images, the longitude and latitude range of the overlapping area can be roughly determined. Further, by combining the number of SAR image pixels and the side-look direction, the longitude and latitude of each pixel of the two images can be determined by interpolation. The locations of overlapping regions in SAR images can be quickly determined by using the longitude and latitude of overlapping areas and the longitude and latitude of pixels. In addition, adjacent data blocks in the same strip can also be used to calculate their respective azimuth pixel GPS time intervals according to their respective GPS time and azimuth points, and quickly determine the number of overlapping azimuth pixels and the location of the overlapping area by using the uniqueness of GPS time.

It should be noted that, since there is a certain error between the longitude and latitude of the four corners of the image and the real position, the range and azimuth pixels of the determined overlapping area can be expanded outward to ensure that the overlapping area of the two SAR images is completely covered.

According to the above analysis, the calculation of overlapping area can be divided into two parts, which are the calculation of the overlapping area in the same strip and the calculation of the overlapping area in the different strips.

In the same-strip case shown in Figure 3a, we firstly use the GPS time of the master SAR image and the slave SAR image to calculate the overlapping time of the two images, then obtain their corresponding azimuth pixel scopes of the two images according to the consistency of overlapping GPS time and overlapping azimuth pixels. In the next step, the

longitude and latitude of the four corners of the two images are interpolated to obtain the overlapping range pixel scopes of the two images. Finally, the overlapping areas of the two images are expanded outward to ensure that the overlapping area of the two SAR images is completely covered.

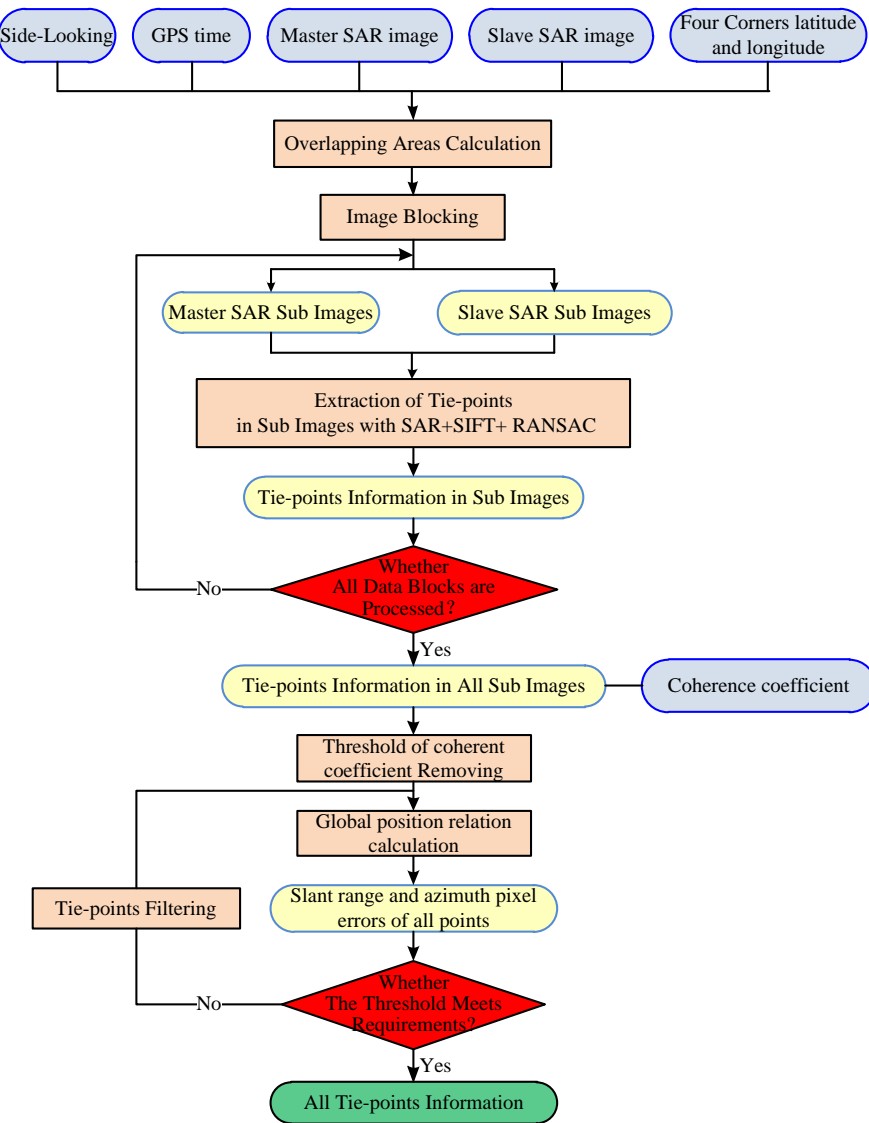

**Figure 2.** Flow chart of automatic extraction of tie-points.

In the different-strip case shown in Figure 3b, the longitude and latitude of the four corners of the two images are firstly interpolated to obtain the overlapping azimuth and range pixel scopes of the two images. Then, the overlapping areas of the two images are expanded outward to ensure that the overlapping area of the two SAR images is completely covered.

### 2.2.2. Image Blocking

In order to ensure that the extraction results of tie-points are evenly distributed in the overlapping area, and prevent too many or too few tie-points in the local area from affecting the subsequent processing, it is necessary to block the overlapping image, select and eliminate the tie-points for each sub image and improve the accuracy of subsequent processing.

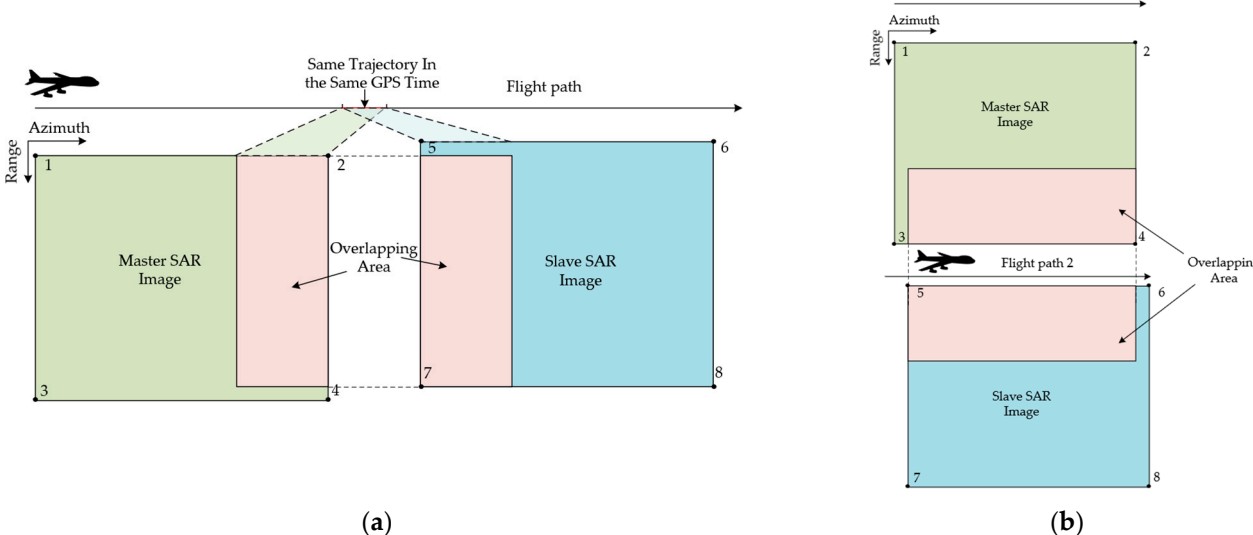

(**a**)     (**b**)

**Figure 3.** Calculation of overlapping area, where (**a**) is the same-strip case and (**b**) is the different-strip case.

The image is segmented in range and azimuth according to the size of the overlapping area. A schematic diagram of image blocking can be seen in Figure 4, where the azimuth pixels of overlapping areas can be divided into m parts and the range pixels of overlapping areas can be divided into n parts. Thus the overlapping areas can be divided into m*n blocks/sub images and each block/sub image has overlapping areas around the other blocks to ensure the continuity of subsequent processing results. The number of n and m is determined by the size of the overlapping area and the size of each block. For airborne millimeter-wave InSAR, the block size of intra/inter strip data is generally set to $512 \times 512$ or $1024 \times 1024$ pixels, and 64 or 128 pixels overlap between adjacent data blocks.

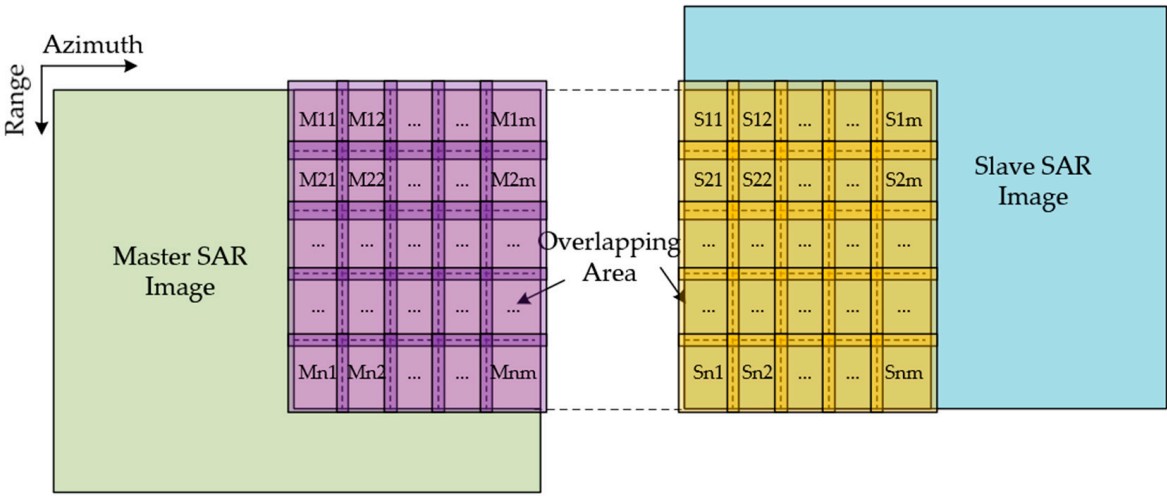

**Figure 4.** Diagram of image blocking.

### 2.2.3. Extraction of Tie-Points in Sub Images

The method of tie-points extraction based on SIFT + RANSAC has been widely used in optical image matching [19,20]. In recent years, this method has also been used in SAR image tie-points extraction, and the SAR-SIFT algorithm, which is more suitable for SAR image tie-point extraction, has been derived [21,22,34]. Therefore, this paper selects the algorithm based on SAR-SIFT-RANSAC to extract the tie-points from each pair of approximately identical SAR images. The diagram of the SAR-SIFT-RANSAC algorithm can be seen in Figure 5.

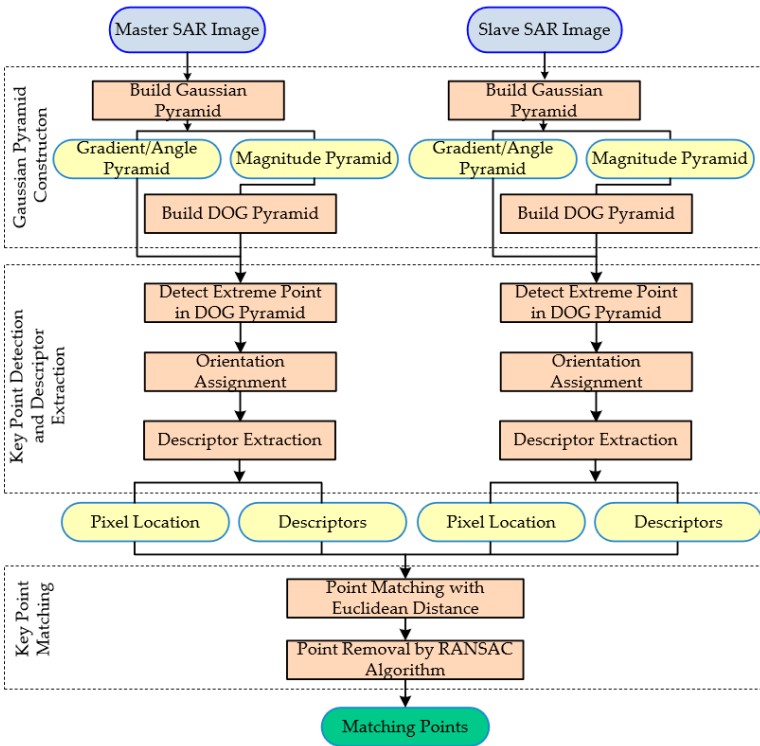

**Figure 5.** Diagram of SAR + SIFT + RANSAC processing flow.

The processing flow of the SAR-SIFT-RANSAC algorithm mainly includes Gaussian pyramid construction, key point detection and descriptor extraction and key point matching. The following is a brief illustration of the above process.

(a)   Gaussian pyramid construction

The Gaussian image pyramid is first constructed by convolving the image with Gaussian filters at different scales (octaves). The octave original image is obtained by multilevel downsampling from original master/slave SAR images. Additionally, each octave consists of different intervals with different Gaussian ambiguity coefficients. The image $L(x, y, \sigma)$ in the Gaussian pyramid can be expressed as follows:

$$L(x, y, \sigma) = G(x, y, \sigma) \otimes I(x, y) \tag{7}$$

$$G(x, y, \sigma) = \frac{1}{2\pi\sigma^2} e^{-\frac{x^2 + y^2}{\sigma^2}} \tag{8}$$

$$\sigma(o, r) = \sigma_0 \cdot 2^{o + \frac{r}{s}} \quad o \in [0, \ldots, O-1], r \in [0, \ldots, s+2] \tag{9}$$

where $\sigma_0$ is Gaussian ambiguity original value, $\sigma$ is Gaussian ambiguity value in different pyramid octaves and intervals, $O$ is the number of pyramid octaves, $s$ is the number of intervals in each octave, $L$ is the magnitude pyramid SAR images, $x$ and $y$ are the horizontal and vertical location in images and $\otimes$ is the convolution operator.

At the same time, the gradient pyramid and angle pyramid, which is the difference from the traditional SIFT algorithm, can be obtained as follows [34]:

$$L_\sigma^{o,r} = \sqrt{\left(L_{x,\sigma}^{o,r}\right)^2 + \left(L_{y,\sigma}^{o,r}\right)^2} \tag{10}$$

$$R_\sigma^{o,r+1} = \arctan\left(\frac{L_{y,\sigma}^{o,r+1}}{L_{x,\sigma}^{o,r+1}}\right) \tag{11}$$

$$L_\sigma^{o,r+1} = \sqrt{\left(L_{x,\sigma}^{o,r+1}\right)^2 + \left(L_{y,\sigma}^{o,r+1}\right)^2} \tag{12}$$

where $L_\sigma^{o,r}$ is the gradient value in the pyramid of the $o$th octave and $r$th interval, $r \in [0, \dots, s + 1]$, $R_\sigma^{o,r+1}$ is the angle value in the pyramid angle of the $o$th octave and $(r + 1)$th interval, and $L_{x,\sigma}^{o,r+1}$ and $L_{y,\sigma}^{o,r+1}$ are the horizontal and vertical derivatives of gradient magnitude image $L(x, y, \sigma)$ in $o$th octave and $r$th interval.

Then, a series of difference of Gaussian (DoG) images is obtained by subtracting adjacent Gaussian pyramid images.

(b)    Key point detection and descriptor extraction

Extreme points are discrete and detected in the Gaussian pyramid images. Therefore, the position and scale of key points are precisely determined by three-dimensional quadratic function fitting, while the key points with low contrast and unstable edge response points are removed (because the DoG operator will produce a strong edge response) so as to enhance the matching stability and improve the anti-noise ability.

Dominant orientations are assigned to key points to maintain rotation invariance. Both the dominant orientation and descriptor extraction are based on the gradient orientation histogram in a scale-dependent neighborhood [34]. The peak value of the orientation histogram represents the direction of the neighborhood gradient at the feature point, and the maximum value in the histogram is the main direction of the key point. At this point, there are three pieces of information for each key point: location, scale and direction.

Instead of using a square neighborhood and $4 \times 4$ square sectors as in the original SIFT descriptor, a GLOH-like circular neighborhood with a radius of $12\sigma$ and log-polar sectors (17 location bins) is utilized to create a feature descriptor, and a series of experiments show that GLOH obtains the best results [22]. Additionally, 128 elements in one descriptor are used to describe the characteristics of this key point [34].

Additionally, [34] and experimental results show that the more DoG image intervals there are, the more points are picked, which means that more points can participate in the matching of master and slave images, and there is a higher chance of selecting more tie-points.

(c)    Key point matching

Based on the above steps, there are four pieces of information for each key point: location, scale, direction and descriptor.

We can obtain the Euclidean distance by using the descriptors of key points and remove inappropriate points according to the threshold value. For each point selected by the master image, the Euclidean distance between it and the points selected by the slave image is calculated, and the point with the minimum distance is selected as the matching point of the slave image [22].

Finally, the locations of the selected key point pairs are used in the RANSAC algorithm to remove outliers. Since the RANSAC method is a mature algorithm [21], it will not be described in detail in this article.

Based on the above processing flow, the tie-points of master and slave images can be effectively extracted. In the same strip, there is little difference between the characteristics of the key points in the master image and the slave image, so it is easier to extract and match the key points. At this time, the number of selected DoG image intervals is relatively small, generally 3. However, in different strips, the geometric characteristics and amplitude characteristics of the tie-points are relatively different in the master and slave images due to the difference in radar nadir angle, so it is relatively difficult to extract and match the key points. At this time, the selected DoG image intervals need to be expanded to obtain more key points for point extraction and matching so as to improve the matching success rate of high points and obtain more connection points.

Therefore, due to the different degrees of difficulty in extracting the tie-points within and between strips, the methods selected for processing the two are slightly different. The selection of tie-points in data blocks between strips requires a higher number of DOG pyramid intervals and a longer processing time, which is important for tie-points extraction.

### 2.2.4. Threshold of Coherence Coefficient Removing

The SAR-SIFT-RANSAC-based algorithm uses the texture structure of the image to extract the tie-points, and some of the tie-points are in low coherence positions. The coherence coefficient of tie-points is positively related to the phase and the elevation inversion accuracy, so it is necessary to calculate the coherence coefficient of each tie-point. The coherence coefficient is calculated according to the position of each tie-point in the image, and the points with low coherence are eliminated to improve the accuracy of elevation inversion.

The coherence coefficient of tie-points can be calculated using the single look complex (SLC) image pairs:

$$\gamma = \frac{\left| \sum\limits_{i=1}^{M} \sum\limits_{j=1}^{N} s_{A_1}(i,j) s_{A_2}^*(i,j) \right|}{\sqrt{\sum\limits_{i=1}^{M} \sum\limits_{j=1}^{N} \left\| s_{A_1}(i,j) \right\|^2 \sum\limits_{i=1}^{M} \sum\limits_{j=1}^{N} \left\| s_{A_2}(i,j) \right\|^2}} \tag{13}$$

where $s_{A_1}$ is the SLC data of antenna $A_1$, $s_{A_2}$ is the SLC data of antenna $A_2$, $(i, j)$ is the pixel location of the point in the SLC image, numeric attributes of $i$ and $j$ are integers, And $M$ and $N$ are the window of coefficient calculation. For airborne millimeter-wave InSAR, the size of $M$ and $N$ can be set to 16 or 32 pixels.

The pixel position of tie-points is generally non-integral. The four coherence coefficients are obtained by using four pixels around the tie-point, and then the coherence coefficient at the tie-point can be obtained by bilinear interpolation using the four coherence coefficients.

After the coherence coefficients of tie-points in the master SAR image and slave SAR image are calculated, the elimination of tie-points is carried out by the threshold of coherence coefficient. The tie-point can be reserved if the coherence coefficients of tie-points in the master and slave data are higher than the coherence threshold $\gamma_t$ at the same time. Moreover, the tie-point can be removed if the coherence coefficient of any image (master or slave) is below the threshold value $\gamma_t$. For airborne millimeter-wave InSAR, the value of the coherence threshold $\gamma_t$ can be set to 0.9. The status of tie-points ($S_{tp}$) can be expressed by Equation (14):

$$S_{tp} = \begin{cases} reserved & \gamma_{maser} > \gamma_t, \gamma_{slave} > \gamma_t \\ removed & others \end{cases} \tag{14}$$

### 2.2.5. Filtering of Tie-Points

Each pair of tie-points can establish the following relationship in the sub-image:

$$x_2 = a_0 + a_1 x_1 + a_2 y_1 + a_3 x_1 y_1 + a_4 x_1^2 + a_5 y_1^2 \tag{15}$$

$$y_2 = b_0 + b_1 x_1 + b_2 y_1 + b_3 x_1 y_1 + b_4 x_1^2 + b_5 y_1^2 \tag{16}$$

where $(x_1, y_1)$ is the pixel of master SAR image and $(x_2, y_2)$ is the pixel of slave SAR image.

Each tie-point pair is used to solve the polynomial coefficients ($a_0, a_1, a_2, a_3, a_4, a_5, b_0, b_1, b_2, b_3, b_4, b_5$) in Equations (15) and (16) by solution of linear equations.

Then, the pixels of tie-points in the master image and the coefficients are used to fit the theoretical pixel position $(\hat{x}_2, \hat{y}_2)$ in the slave SAR image.

The matching error (range matching error $e_{ran}$ and azimuth matching error $e_{azi}$) can be expressed as follows:

$$e_{ran} = \hat{x}_2 - x_2 \tag{17}$$

$$e_{azi} = \hat{y}_2 - y_2 \tag{18}$$

The pixel position of tie-points in sub images may have a uniform distribution in their sub images, but they are not uniform in the whole overlapping SAR images. Although the

matching error may have a minimum in the sub images, they cannot reach minimum in the whole overlapping SAR images. Moreover, the tie-points selected for each data block based on the SAR-SIFT-RANSAC algorithm are local optimal solutions, but they are not globally optimal for the whole SAR image.

Therefore, it is necessary to recalculate the position relationship of the tie-points in two images to obtain the range and azimuth pixel error of each pair of tie-points, and the processing flow is shown in Figure 6.

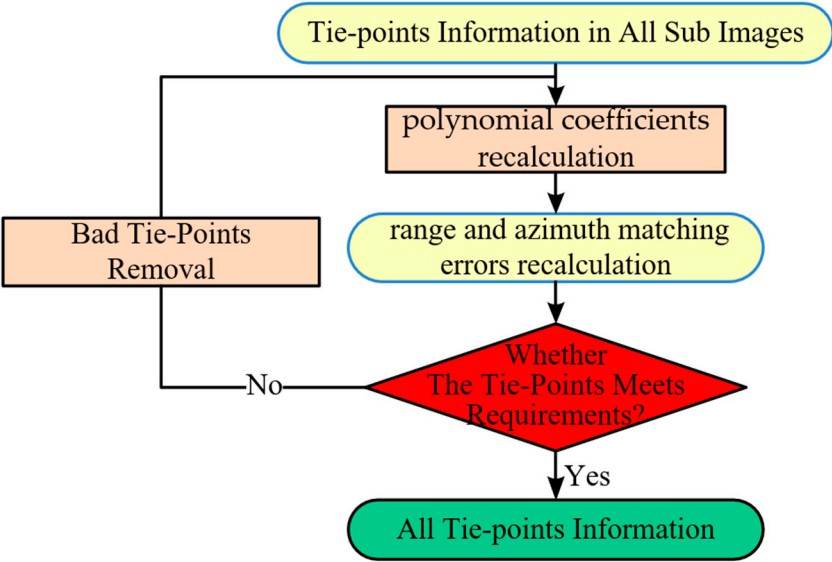

**Figure 6.** Diagram of filtering of tie-points.

That is, the polynomial coefficients ($a_0$, $a_1$, $a_2$, $a_3$, $a_4$, $a_5$, $b_0$, $b_1$, $b_2$, $b_3$, $b_4$, $b_5$) are firstly obtained by polynomial fitting using the tie-points in the whole overlapping SAR images. Then, the matching error in the whole overlapping SAR images can be recalculated using Equations (17) and (18). At this time, the matching errors of some tie-points considered to be optimal in sub images may behave poorly in the whole overlapping areas.

The tie-points with large pixel errors are considered to be unsuccessful in matching and need to be eliminated. In general, the range pixel error and azimuth pixel error of tie-points within/between strips should be less than 1 pixel. If there are relatively few tie-points in two adjacent SAR images, the threshold values of range and azimuth pixel errors of tie-points between strips can be extended to 2 pixels.

After the tie-points are eliminated based on the pixel error threshold, the position relationship needs to be recalculated, which means the polynomial coefficients ($a_0$, $a_1$, $a_2$, $a_3$, $a_4$, $a_5$, $b_0$, $b_1$, $b_2$, $b_3$, $b_4$, $b_5$) and matching errors need to be recalculated using Equations (15)–(18). Thereafter, the quality of tie-points can be evaluated by their matching errors.

If the pixel matching error still does not meet the threshold requirements, iterative operation is required until the tie-points all meet the requirements of the range and azimuth pixel error threshold.

### 2.3. Interferometric Calibration Based on Tie-Points

The data processing flow of interferometric calibration based on tie-points is shown in Figure 7. First, the single look complex (SLC) data of the master and slave images are used for interferometric processing to obtain the coherence coefficient and phase unwrapping data. Additionally, the SLC data of the master and slave images can be obtained and accurately registered using high-precision SAR processing procedures, which include focused imaging processing and high-precision motion compensation in the SAR imaging model of two antennas.

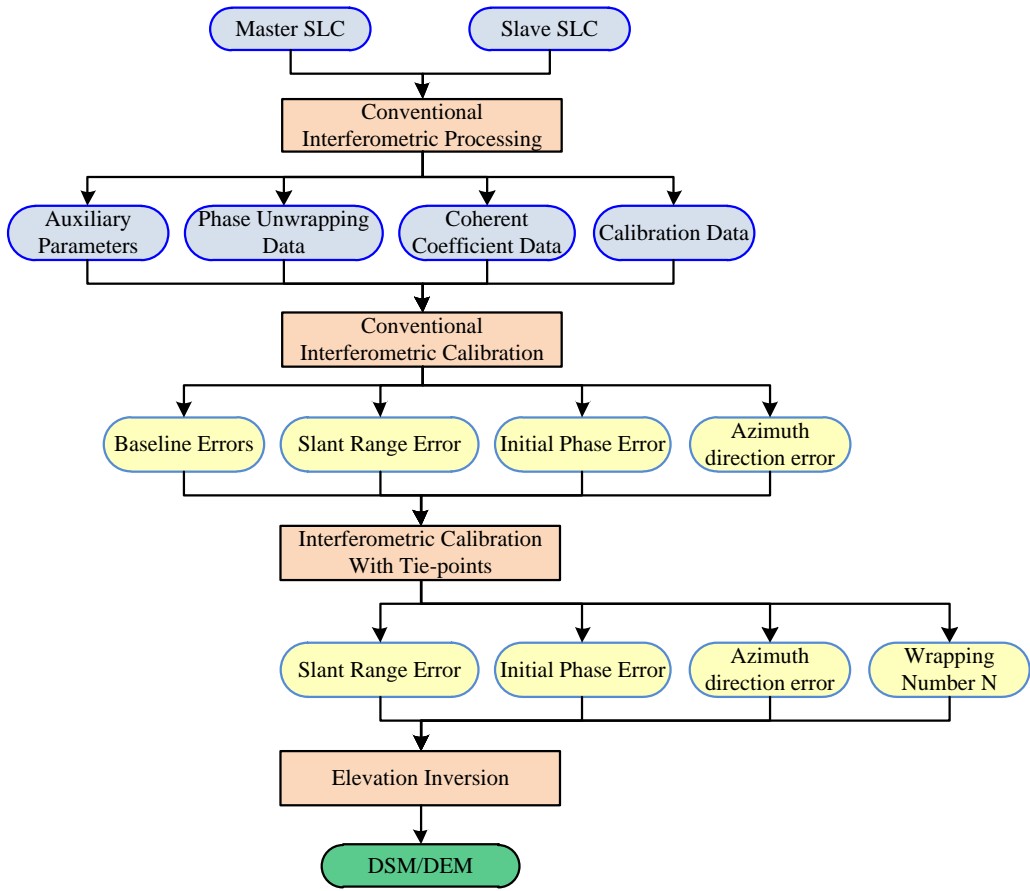

**Figure 7.** Data processing flow chart of interferometric calibration based on tie-points.

Then, combined with ground control points and imaging auxiliary parameters, conventional interferometric calibration processing is carried out to obtain the baseline error, slant range error, initial phase error and azimuth error in planned calibration areas in which the sensitivity equation was constructed and the error was obtained through pseudo-inverse solution in calibration method [13]. Additionally, the procedure is only carried out in a specific and small calibration area where a column of calibration ground points can be arranged evenly along the range orientation of the SAR image. Moreover, this step is mainly to obtain InSAR system error (baseline error) and other parameter original errors (slant range error, initial phase error and azimuth error). Moreover, the InSAR system error is available in all different flights if the InSAR radar is not removed from the aircraft. Other parameter original errors are different in different flights, and only used in the current flight. Furthermore, the phase N value, which is the 2π wrapping number, is only useful in the block data that is calibrated with ground points. That is, in the same flight, the phase N values of nearly all data blocks need to be solved. Therefore, despite the interference calibration process, the phase N value still needs to be solved in multiple sorties, and other parameter errors (slant range error, initial phase error and azimuth error) still need to be solved in other sorties. The solution of this condition requires a large number of ground control points, but the layout of a large number of control points will, of course, bring a lot of costs, and some places are not even suitable for calibration, especially in mountainous and urban areas. In this case, it is necessary to use the tie-points to process the data of multiple sorties and recalibrate the error parameters.

In the next step, interferometric calibration based on tie-points is carried out using the tie-points' information, which is the main process of the processing flow. Based on the premise that it is assumed that the height errors (the actual real height minus InSAR inversion height) of the tie-points are consistent in the master and slave images, the method uses the known height information of the master image to recalibrate the parameter

errors (slant range error, initial phase error and azimuth error) and the phase N value. Furthermore, the slant range error can be calculated by the subtraction of ideal slant range and the modulation of the slant range in images using the known tie-point in slave images and atmospheric compensation operator. Then, the points where the slant range error is higher than the threshold (generally 3*sigma of the slant range error) will be removed to guarantee the accuracy of the slant range error. At the same time, the azimuth error can be solved in the same way using the known height and plane information of the master image. Moreover, based on the determined baseline error, slant range error and azimuth error, the theoretical interferometric phase can be calculated using the slant range difference between the master antenna and slave antenna. The initial phase error and the phase N value can be obtained using the phase difference between the theoretical interferometric phase and unwrapping phase. In the same way, the points where the initial phase error is higher than the threshold (generally 3*sigma of the initial phase error) will be removed to guarantee the minimum height error between the tie-points in the master and slave images. The following is a relevant description of the main processing.

Moreover, for the experimental area without ground control points, it is necessary to use the tie-points to coordinate the interferometric calibration and recalibrate the error parameters. For the same sortie, the flight state is relatively stable, so it can be considered that error compensation parameters, such as slant range error, initial phase error and azimuth range error, are consistent. After the above parameters are determined, the block adjustment is carried out by using the tie-points to calibrate the phase N value, which is the $2\pi$ wrapping number, between data blocks.

For different flights, the interferometric calibration error parameters vary greatly between different sorties due to the influence of flight environment, radar state difference when turned on, etc. Therefore, it is necessary to calibrate the slant range error, initial phase error and azimuth range error by using the tie-points in the overlapping area between different sorties through block adjustment, while the calibration of the phase N value is the same as the processing method of the same sortie. The height of the same tie-points in the known elevation data block is used as the calibration point to calibrate the adjacent data block and perform elevation inversion. Additionally, the slant range error $\Delta R_{slant}$, initial phase error $\phi_{initial}$, azimuth range error $\Delta d_{azi}$ and the phase N value can be expressed as follows:

$$\Delta R_{slant} = R_{ideal} - f\left(R_{image}\right) \tag{19}$$

$$\Delta d_{azi} = d_{azi\_ideal} - d_{azi\_image} \tag{20}$$

$$\Delta\phi = \phi_{ideal} - \phi_{uwp} = 2\pi \bullet N + \phi_{initial} \tag{21}$$

where $R_{ideal}$ is the ideal slant range of calibrating tie-points and calculated using known elevation and plane information of tie-points, $R_{image}$ is the image slant range, $f(\bullet)$ is the atmospheric compensation operator, which can be designed for an empirical model or poly-fit model, $d_{azi\_ideal}$ is the ideal position of calibrating tie-points and calculated using known elevation and plane information of tie-points and $d_{azi\_image}$ is the image azimuth position of calibrating tie-points. Additionally, $\Delta d_{azi}$ can influence the plane information of tie-points and, thereafter, affect the elevation of subsequent calibration points. $\phi_{ideal}$ is the ideal interferometric phase of calibrating tie-points and calculated using known elevation and plane information of tie-points and $\phi_{uwp}$ is the unwrapping phase of calibrating tie-points. $\phi_{initial}$ can be obtained using the wrapping operator $Wrap(\bullet)$. The wrapping operator $Wrap(\bullet)$ can be expressed as follows:

$$\phi_{initial} = Wrap(\Delta\phi) = \mathrm{mod}(\Delta\phi + \pi, 2\pi) - \pi \tag{22}$$

where $\mathrm{mod}(\bullet)$ is the modulo operator.

The phase N value can be easily obtained using Equation (21). When all error parameters are determined, the elevation data and corresponding products can be obtained through elevation inversion.

Comparing the two calibration modes, the calibration based on ground points uses the height of the ground control point as the true value for calibration, while the calibration based on tie-points uses the known height and plane information to carry out elevation transfer and error parameter calibration under the assumption that the height errors (the actual real height minus InSAR inversion height) of the tie-points are consistent in the master and slave images. The main goal of the calibration based on ground points is to obtain the InSAR system baseline error, while the main goal of the calibration is to obtain other parameter errors (slant range error, initial phase error and azimuth error) in different sorties and the phase N value in all data blocks of all sorties.

Moreover, the calibration based on ground control points uses a partial differential equation to construct the sensitivity equation of ground points and obtain the parameter errors through pseudo-inverse solution of the sensitivity equation, while the calibration based on tie-points uses the height transfer mode to calibrate the parameter errors (slant range error, initial phase error and azimuth error) of slave images by geometric model solving and threshold elimination. The variance in slant range error, phase error and azimuth error of the ground points is relatively small because of the centimeter error measurement of the ground point, while the variance in slant range error, phase error and azimuth error of the tie-points is relatively expanded because of the relatively large height transfer error and the position error of the tie-points, which is necessary to obtain accurate errors through threshold elimination. In addition, the mode of calibration based on ground control points just needs to spread the ground control points evenly along the slant range orientation, while that of calibration based on tie-points needs to spread the tie-points evenly among the overlapping areas between the master images and the slave images to guarantee the accuracy of other parameter errors (slant range error, initial phase error and azimuth error). That is, the uniform selection of connection points is necessary, and our proposal involves just enough to produce uniform tie-points.

Finally, using the interferometric data and these error parameters, the elevation data of the calibration field can be obtained through elevation inversion, and digital surface model (DSM) and digital elevation model (DEM) products can be obtained through further processing [35].

## 3. Results

### 3.1. Automatic Extraction of Tie-Points

The experiment of tie-point extraction of adjacent data blocks within/between strips is carried out on airborne millimeter-wave InSAR data using the automatic tie-point extraction method described in this paper. The measured data are the airborne millimeter-wave dual-baseline InSAR system [32], and this airborne system has two baselines and its imaging resolution is 0.3 m. The InSAR 3D products of the system meet the 1:5000 scale requirement for topographic mapping in difficult areas of China. Table 1 shows the system parameters of airborne millimeter-wave dual-baseline InSAR. The experimental results fully verify the feasibility and effectiveness of the method proposed in this paper.

#### 3.1.1. Automatic Extraction of Tie-Points in One Flight Strip

The tie-points of two adjacent data blocks in the same strip were extracted by using the data acquired by airborne millimeter-wave radar. The adjacent SAR images and the extracted tie-points are shown in Figure 8. The number of range and azimuth pixels of the master image and slave image are 8704 and 9712, respectively, and the range and azimuth pixels of the overlapping area of the two images are about 8500 × 2600. The green crosses in the figure are the automatically extracted tie-points.

The statistical results show that the number of tie-points extracted from the two images is 124, and the tie-points are evenly distributed in the overlapping area of the images. Figure 9 shows the comparison results of 6 pairs of randomly selected tie-points and surrounding SAR images. It can be seen from the figure that the texture and gray value

of the master and slave SAR images around the tie-points in the strip are consistent, which is conducive to the matching of tie-points.

**Table 1.** Experimental parameters of the airborne millimeter-wave InSAR system.

| Parameters | Value |
|---|---|
| Band | Ka |
| Frequency/GHz | 35 |
| Chirp bandwidth/MHz | 900 |
| Longer baseline/m | 0.313 |
| Shorter baseline/m | 0.087 |
| Baseline inclination/deg | 45 |
| Carrier height/m | 3000 |
| Nadir angle/deg | 38~52 |
| Range resolution/m | 0.3 |
| Azimuth resolution/m | 0.3 |
| Average flight speed/(m/s) | 65 |

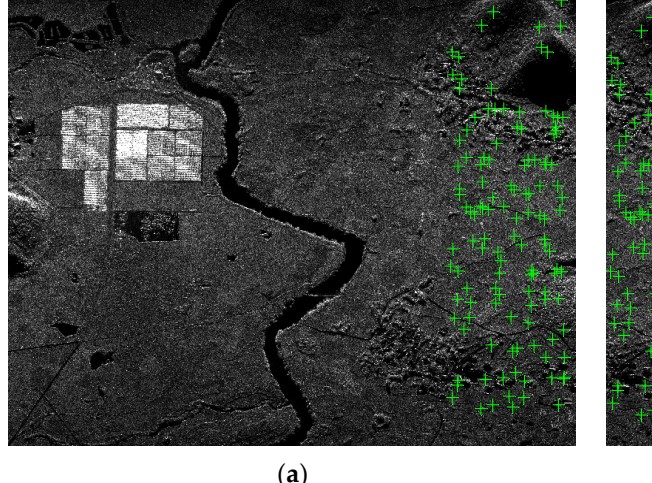 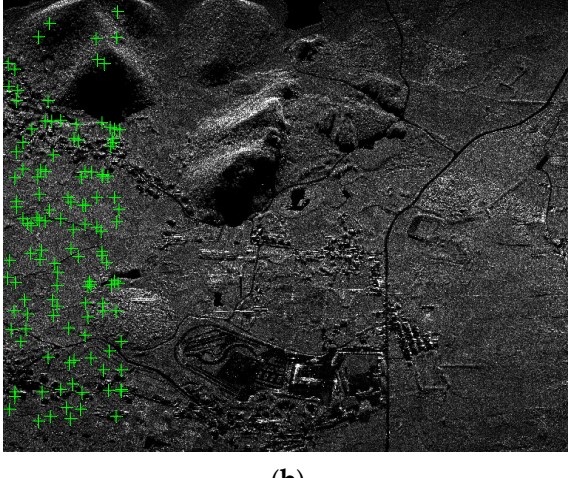

(**a**)  (**b**)

**Figure 8.** The position of the tie-points between the master image and the slave image within the strip, where (**a**) is the master image and (**b**) is the slave image. The horizontal axes represent azimuth direction, and vertical axes represent the slant range direction.

Further, this paper analyzes the coherence coefficient of each tie-point, and the results are shown in Figure 10. The abscissa in the figure is the number of tie-points, and the ordinate is the coherence coefficient. It can be seen from the figure that the average coherence of the master and slave images is about 0.975, indicating that the tie-points extracted in this paper have high coherence. The coherence objectively reflects the stability of phase error and the accuracy of elevation inversion, and ensures the accuracy of elevation inversion of subsequent tie-points points.

After the tie-point automatic selection step, the range pixels and azimuth pixels of each tie-point have been already obtained. Additionally, we can use Equations (15) and (16) to solve the polynomial coefficients ($a_0$, $a_1$, $a_2$, $a_3$, $a_4$, $a_5$, $b_0$, $b_1$, $b_2$, $b_3$, $b_4$, $b_5$) and obtain the theoretical range and azimuth pixel position of the slave SAR images. The matching error is obtained by subtracting the theoretical pixel position from the actual pixel position, shown in Equations (17) and (18). The matching accuracy of the tie-points within the strip is analyzed quantitatively, and the range and azimuth matching errors of the tie-points are shown in Figure 11a,b, respectively. The abscissa in the figure is the number of tie-points,

and the ordinate is the matching error. It can be seen from the figure that the range and azimuth matching errors of the 124 tie-points are within 0.5 pixels, which conforms to the requirements of the pixel error threshold of the tie-point, and fully demonstrates the reliability of the method in the extraction of tie-points in the same strip.

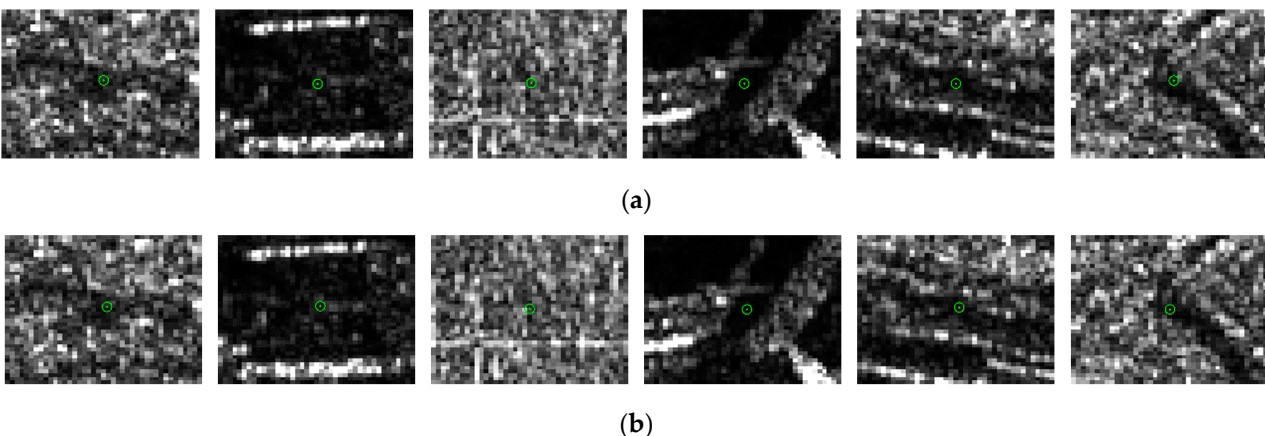

(**a**)

(**b**)

**Figure 9.** Comparison of six pairs of randomly selected tie-points in the strip and their SAR images, where (**a**) is the master image and (**b**) is the slave image.

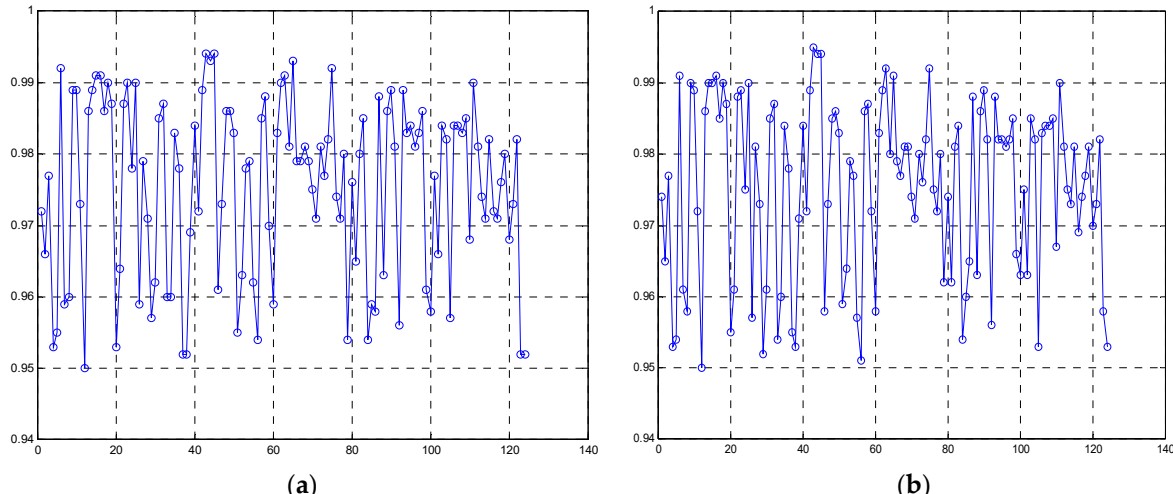

(**a**)                                     (**b**)

**Figure 10.** Coherence coefficient of the tie-points in the master and slave SAR images within strips, where (**a**) is the master image and (**b**) is the slave image.

### 3.1.2. Automatic Extraction of Tie-Points between Flight Strips

Similarly, this paper uses airborne millimeter radar data to extract the tie-points of two adjacent data blocks between strips. The SAR image and the extracted tie-points are shown in Figure 12. The range and azimuth pixels of the two SAR images are 8704 and 9712, respectively. The range and the azimuth pixels of the overlapping area are about $4000 \times 9500$. The green crosses in the figure are the automatically extracted tie-points.

There are 74 tie-points extracted from the two images, which are evenly distributed in the overlapping area between images. Figure 13 shows the comparison results of six pairs of randomly selected tie-points and surrounding SAR images. It can be seen from the figure that the texture and gray value of SAR images near the tie-points between strips are more different than those within strips, which increases the difficulty of tie-point registration. In the real data processing experiments, the matching of tie-points can be realized by increasing the number of pyramid layers.

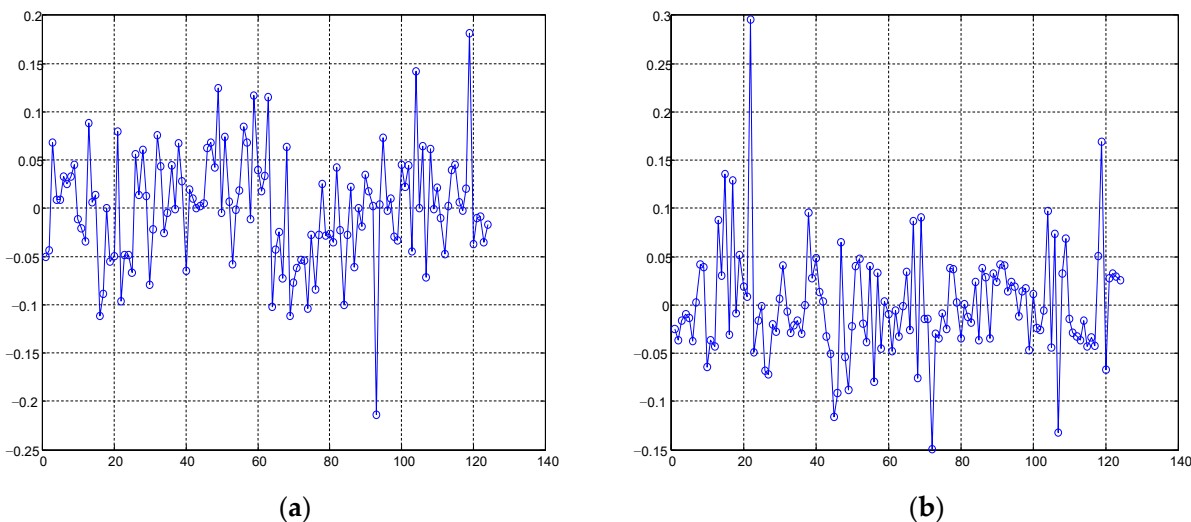

**(a)**　　　　　　　　　　　　　　**(b)**

**Figure 11.** Matching error of the tie-points in the strip, where (**a**) is the range direction and (**b**) is the azimuth direction.

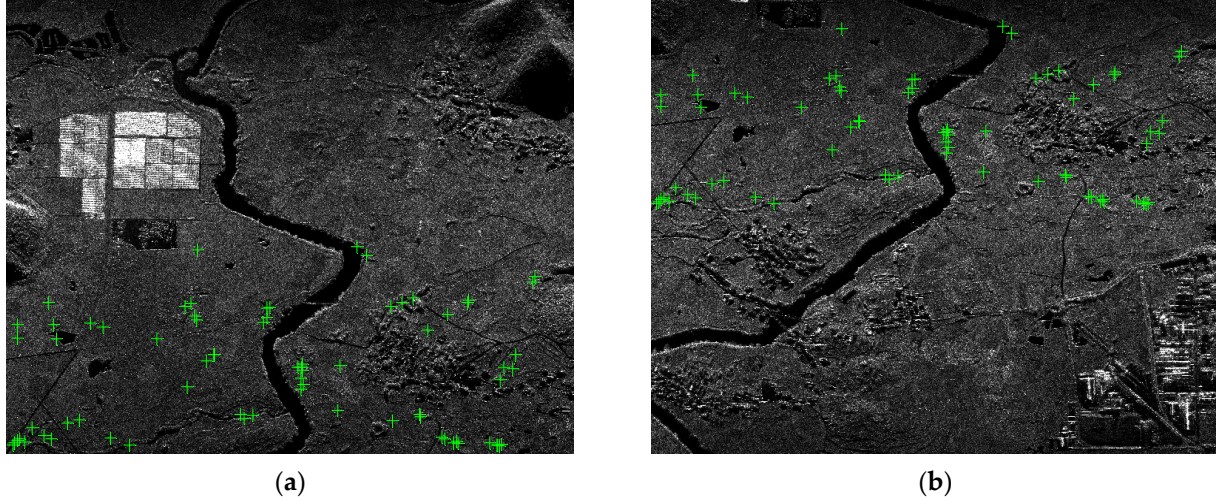

**(a)**　　　　　　　　　　　　　　**(b)**

**Figure 12.** Location of tie-points in the adjacent SAR images between strips, where (**a**) is the master image and (**b**) is the slave image. The horizontal axes represent the azimuth direction, and vertical axes represent the slant range direction.

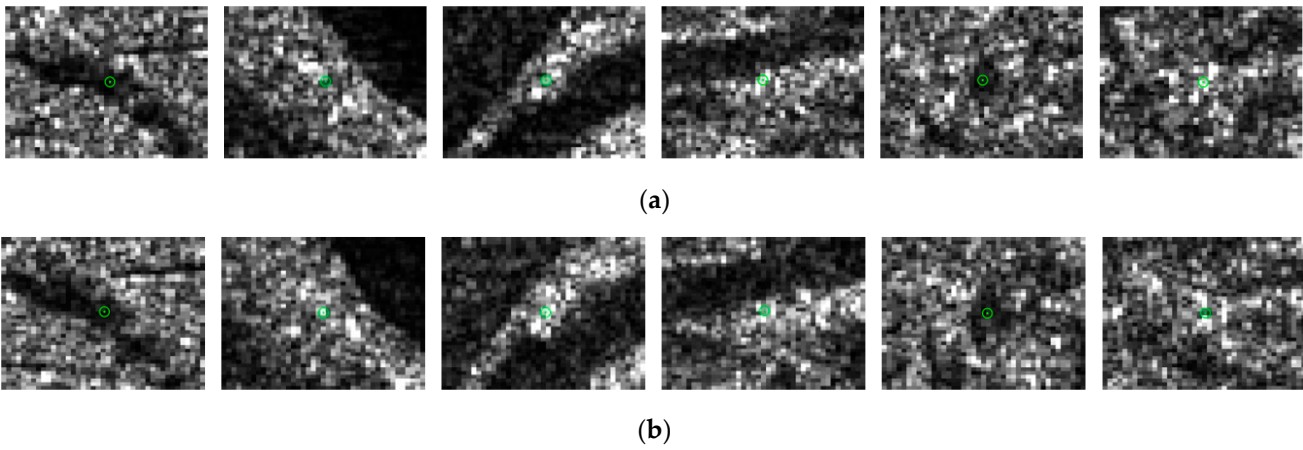

**Figure 13.** Comparison of six pairs of randomly selected tie-points between strips and their SAR images, where (**a**) is the master image and (**b**) is the slave image.

The coherence coefficient of each tie-point was calculated, and the results are shown in Figure 14. The abscissa in the figure is the number of tie-points, and the ordinate is the coherence coefficient value of tie-points. According to statistical analysis, the mean coherence coefficient of the tie-points in the master image is 0.976, and the mean coherence coefficient of the slave image is 0.98. The tie-points extracted between strips also have high coherence, which can support interferometric calibration based on tie-points and subsequent elevation inversion.

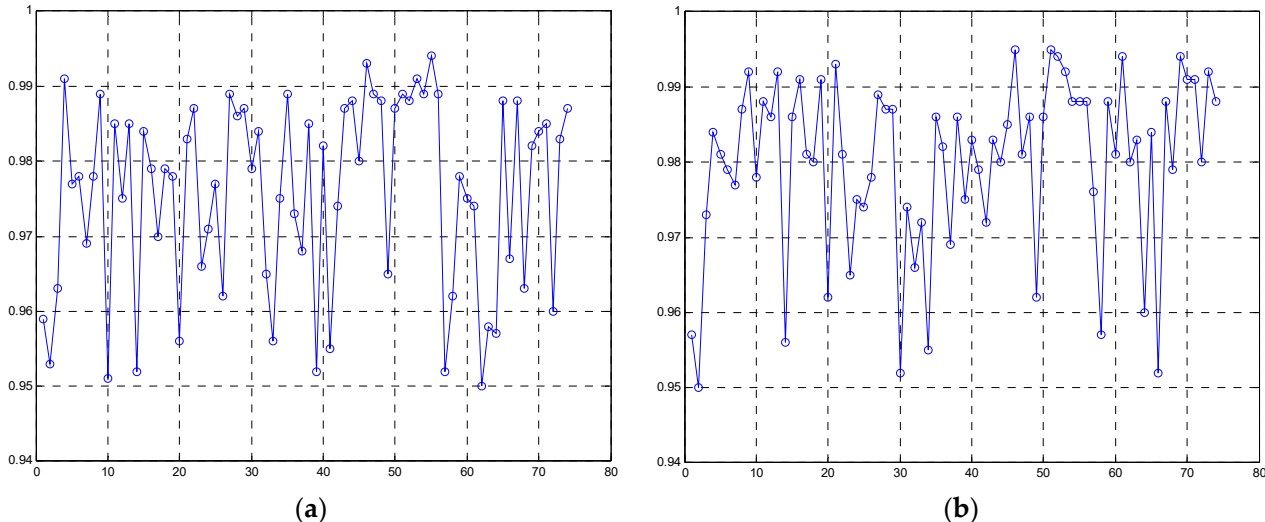

**Figure 14.** Coherence coefficient of the tie-points in the master and slave SAR images between strips, where (**a**) is the master image and (**b**) is the slave image.

After the tie-point automatic selection step, the range pixels and azimuth pixels of each tie-points have been already obtained. Additionally, we can use Equations (15) and (16) to solve the polynomial coefficients ($a_0$, $a_1$, $a_2$, $a_3$, $a_4$, $a_5$, $b_0$, $b_1$, $b_2$, $b_3$, $b_4$, $b_5$) and obtain the theoretical range and azimuth pixel position of the slave SAR images. The matching error is obtained by subtracting the theoretical pixel position from the actual pixel position, shown in Equations (17) and (18). The matching accuracy of tie-points between strips is analyzed quantitatively, and the matching error is shown in the figure below. The abscissa in Figure 15 is the number of tie-points, and the ordinate is the matching error. It can be found from the figure that the range and azimuth matching errors are within 1 pixel. The average pixel error in the range direction is −0.02 and the variance is 0.45, and the average pixel error in the azimuth direction is 0.01 and the variance is 0.57. The analysis of matching error fully shows the reliability of the extraction of tie-points between strips.

### 3.2. Interferometric Calibration Based on Tie-Points

#### 3.2.1. Interferometric Calibration Results of the Same Flight Sortie

The interferometric calibration parameters of the same sortie are relatively stable, so it is only necessary to use the automatically extracted tie-points to calculate the interferometric phase N value between data blocks. In this section, two different strips in the same sortie are calibrated based on the tie-points, which verifies the effectiveness of this algorithm.

The airborne millimeter-wave radar data obtained from the same sortie are used to carry out interferometric calibration and elevation inversion based on the tie-points. The airborne InSAR system is shown in Table 1. Two flight strips are selected, which are recorded as N510 and N511, each strip is about 90 km long and contains 58 data blocks. The elevation information of each complete strip is retrieved by using the tie-points between a data block with known elevation and other data block. There are 1301 pairs of tie-points in the N510 strip and 1382 pairs of tie-points the N511 strip. Figure 16 shows the distribution of relative elevation difference between tie-points in each strip, and Figure 17 shows the

distribution of average elevation difference of tie-points between adjacent data blocks in each strip.

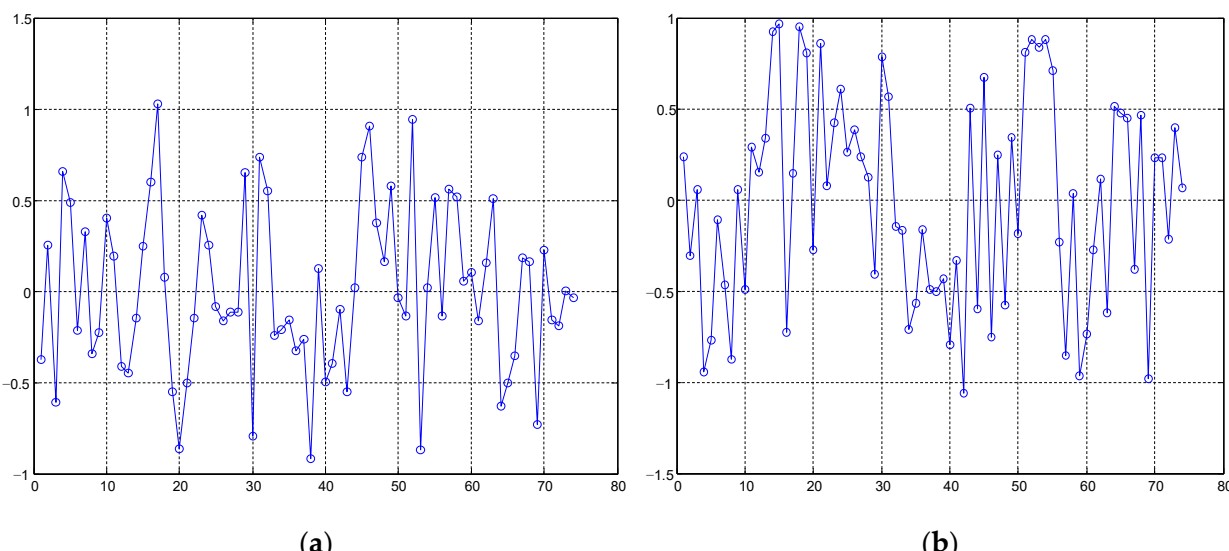

(**a**)            (**b**)

**Figure 15.** Matching error of the tie-points between strips, where (**a**) is the slant range direction and (**b**) is the azimuth direction.

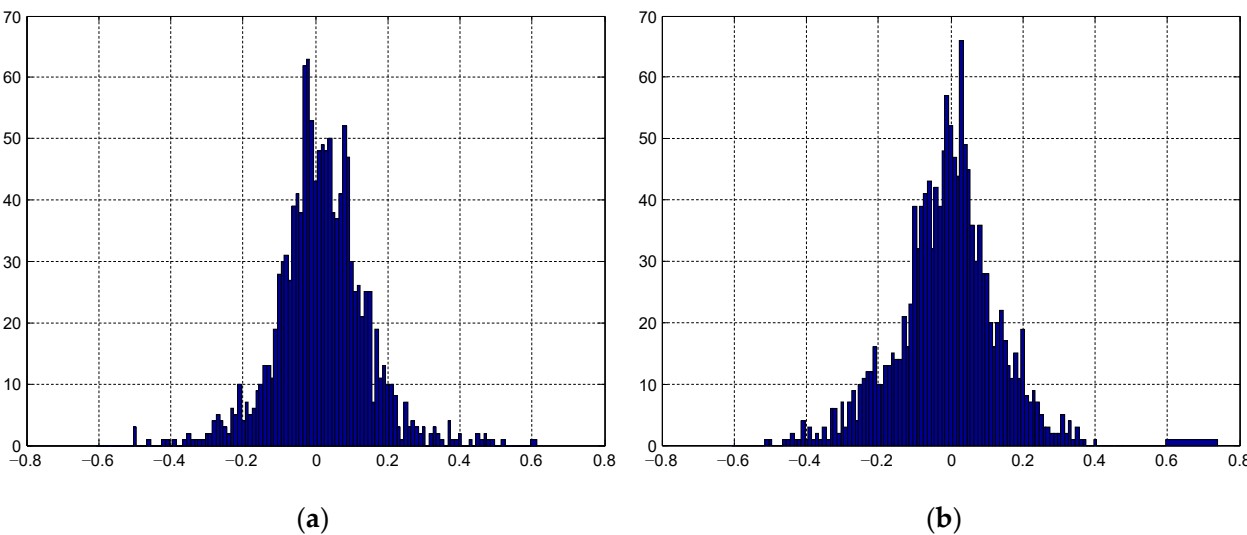

(**a**)            (**b**)

**Figure 16.** The distribution of relative elevation difference between tie-points in each strip, where (**a**) is N510 and (**b**) is N511. The abscissa in the figure is relative elevation difference (in meters) between tie-points, and the ordinate is the number of tie-points corresponding to relative elevation difference.

Figure 16 shows that the relative elevation difference of the tie-points of the two strips is a normal distribution with the mean value of zero. Figure 17 shows that the average relative elevation between adjacent data blocks of each strip fluctuates around zero, indicating that there is no continuous increase or decrease in elevation between adjacent data blocks within each strip, and the elevation fluctuation is small and smooth. In order to further analyze the absolute elevation accuracy of the two strips based on the processing of tie-points, this section selects the control points in the strips to verify the absolute elevation accuracy. Strips N510 and N511 have five and four control points, respectively. The distribution of the difference between the actual elevation of the control point and the elevation obtained by using the calibration of tie-points is shown in

Figure 18. It can be found that the difference between the actual elevation and the inversion elevation of each control point in the two strips is within 0.4, and both meet the high accuracy requirements of large-scale mapping. This shows that the interferometric calibration and elevation inversion method based on the elevation transfer of tie-points can effectively realize the high-precision elevation inversion of different data blocks in the same flight sortie.

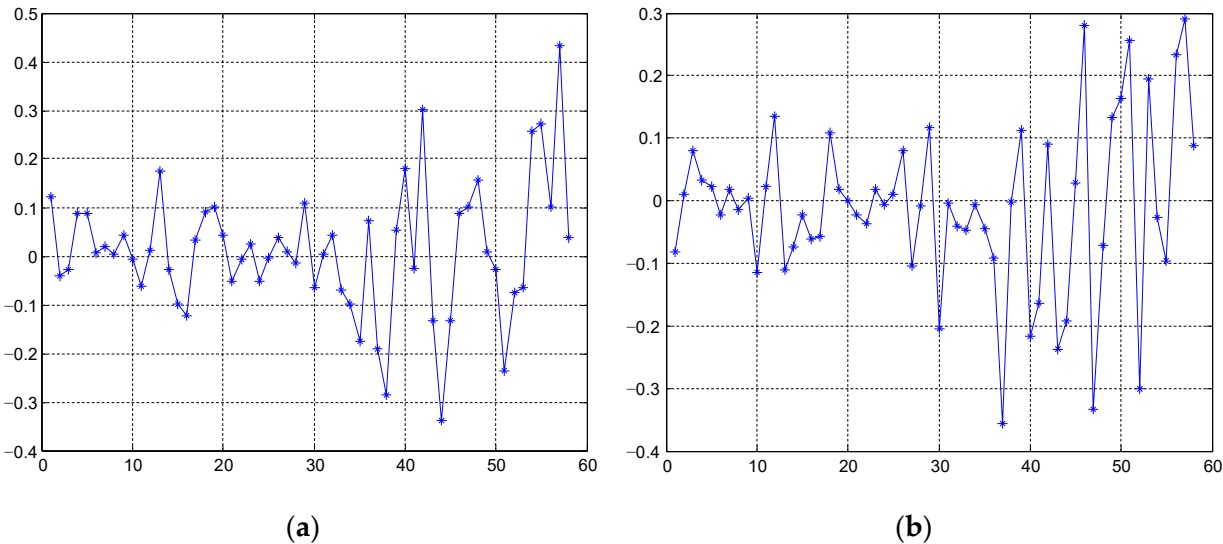

(**a**) (**b**)

**Figure 17.** The distribution of average elevation difference for tie-points between adjacent data blocks in each strip, where (**a**) is N510 and (**b**) is N511. The abscissa in the figure is the number of data blocks within the strip, and the ordinate is average relative elevation difference (in meters) of data blocks.

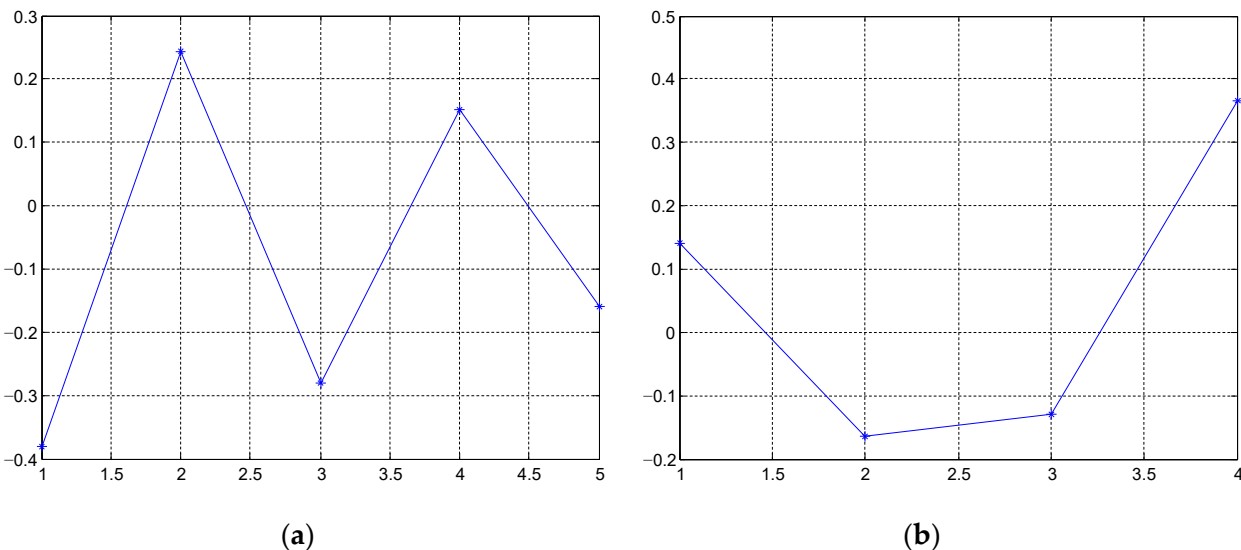

(**a**) (**b**)

**Figure 18.** Difference in measured elevation and inversed elevation of control points in two strips, where (**a**) is N510 and (**b**) is N511. The abscissa in the figure is the number of control points, and the ordinate is absolute elevation difference (in meters) between InSAR elevation of control points obtained by calibration of tie-points and real elevation value of control points.

### 3.2.2. Interferometric Calibration of Different Flight Sorties

Generally, radar equipment will not be disassembled and installed in different flight sorties. Antennas of the airborne millimeter-wave InSAR system are rigidly connected, and the baseline length will not change. Therefore, it can be considered that the baseline length error and baseline inclination error have not changed. However, due to the influence

of weather conditions, different radar startup time and other factors, the slant distance, azimuth time, initial phase, phase unwrapping number and other parameters will change, so it is necessary to recalibrate with the tie-points.

In this subsection, the measured data are used to carry out interferometric calibration and elevation inversion based on the tie-points between strips, and the experimental area with hilly and mountainous terrain is located in Guizhou Province, China. The airborne InSAR system is shown in Table 1. Due to the influence of weather, terrain and airspace, it is difficult to carry out the flight plan and ground control point layout. As a result, there is no ground control point in the experimental area, and 11 adjacent strips are divided into 6 sorties. The distribution of adjacent strips is shown in Figure 19.

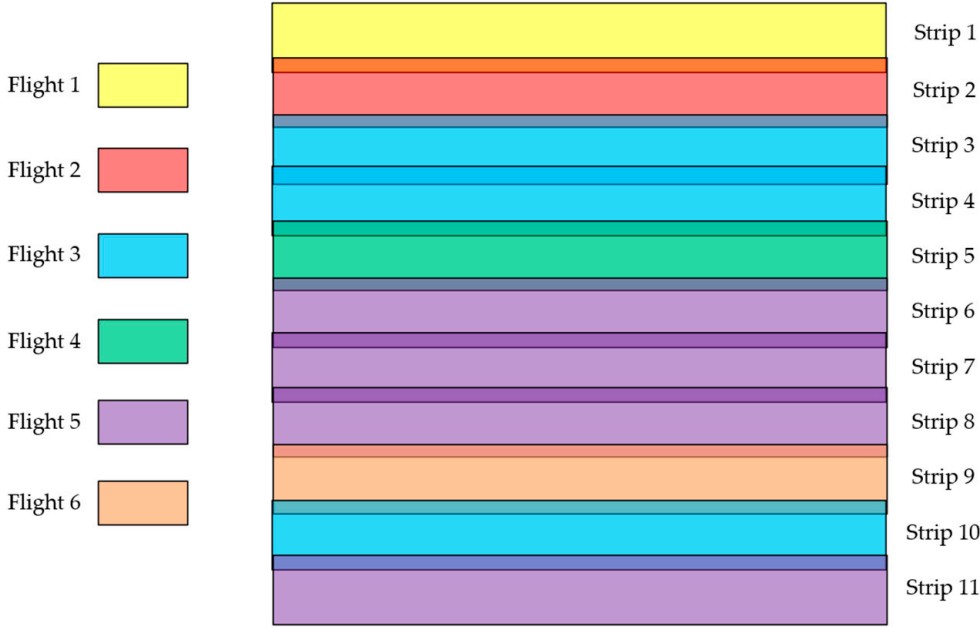

**Figure 19.** Distribution of 11 adjacent strips of 6 different sorties.

In order to evaluate the elevation accuracy, this experiment selects the points with corner reflector characteristics and flat surrounding ground as the marking points according to the SAR image of this area. The marking point is used as the control point to obtain the interferometric calibration parameters, and then the elevation inversion and elevation accuracy inspection are completed. Therefore, the comparison of results in this section is based on the elevation data of marking points. The algorithm in this paper is used to extract the tie-points between different sorties, and to conduct interferometric calibration based on tie-points and elevation inversion. Table 2 shows the comparison between the error parameters obtained based on the interferometric calibration of tie-points and marking points. In Table 2, it is assumed that the calibration parameters and elevation data of strip 1 of the first flight are known, and the parameters of the remaining 10 strips are obtained through interferometric calibration.

Compared with the calibration parameters based on marking points, the calibration parameters of strip 2 based on the tie-points are essentially the same in terms of variation of slant range and azimuth change, and the difference is only on the millimeter scale, while the difference in variation of the initial phase is 0.02 rad, resulting in an elevation error of about 0.3 m. The variation in the initial phase obtained based on the tie-points of strip 3 and strip 4 is essentially the same as the result based on the marking points, resulting in an elevation error of less than 0.1 m. In addition, the difference between variation in slant range and azimuth change are about 0.1 m and 0.4 m, respectively. The azimuth change has little impact on the elevation error, which can meet the requirements of large-scale mapping. The variation in the initial phase of strip 6 and strip 7 is the same, but the difference with strip 8 is 0.03 rad, resulting to an elevation error of about 0.5 m, which may be related to

the quality of the tie-points. In addition, the calibration parameters of strip 6, strip 7 and strip 8 based on marking points are also inconsistent, which may be related to the quality of marking points. The calibration parameters of strip 9 based on tie-points and marking points have little difference, and the difference is within 0.3. The calibration parameters of strip 10 based on tie-points have little difference with the results of strip 3 and strip 4, which verifies the feasibility of error compensation and elevation inversion with the same set of parameters for the same sortie and the effectiveness of the interferometric calibration based on tie-points.

**Table 2.** Interferometric calibration results of 11 adjacent strips of 6 different sorties.

| Interferometric Calibration | Tie-Points Based | | | Marking Points Based | | |
|---|---|---|---|---|---|---|
| Error Parameters | Variation of Slant Range (m) | Variation of Initial Phase (rad) | Azimuth Change (m) | Variation of Slant Distance (m) | Variation of Initial Phase (rad) | Azimuth Change (m) |
| Strip 1, Sortie 1 (Initial, known) | −0.0586 | −0.8866 | 0.5662 | −0.0586 | −0.8866 | 0.5662 |
| Strip 2, Sortie 2 | −0.8656 | −0.7873 | 0.5712 | −0.8622 | −0.7655 | 0.5714 |
| Strip 3, Sortie 3 | −1.5672 | −0.7913 | 0.5781 | −1.4397 | −0.7861 | 0.1786 |
| Strip 4, Sortie 3 | −1.5672 | −0.7913 | 0.5781 | −1.4397 | −0.7915 | −0.1671 |
| Strip 5, Sortie 4 | −2.2536 | −0.7334 | 0.0882 | −1.9001 | −0.7458 | −0.2 |
| Strip 6, Sortie 5 | −1.2106 | −0.7675 | 0.1603 | −0.9040 | −0.7504 | −0.1598 |
| Strip 7, Sortie 5 | −1.1804 | −0.7575 | −0.1674 | −0.9040 | −0.7383 | −0.1671 |
| Strip 8, Sortie 5 | −1.0874 | −0.7298 | −0.4101 | −0.9040 | −0.7671 | −0.1667 |
| Strip 9, Sortie 6 | −0.8667 | −0.8226 | −0.3116 | −0.7449 | −0.8193 | −0.1115 |
| Strip 10, Sortie 3 | −1.6206 | −0.7845 | −0.3417 | −1.4397 | −0.8053 | −0.1408 |
| Strip 11, Sortie 5 | −0.9277 | −0.7236 | −0.1238 | −0.9040 | −0.7363 | −0.1667 |

On the whole, there is little difference between the calibration parameters obtained based on tie-points and marking points for adjacent strips of different sorties. Under the condition that the number of adjacent strips of consecutive different sorties is controllable, the elevation accuracy of 0.5 m can be fully guaranteed. At the same time, the difference in calibration parameters obtained by using the tie-points for different strips of the same sortie is small, and only the slant range difference between strips 7, 8 and 11 of sortie 5 reaches 0.2 m, which is still acceptable. In order to further compare the elevation accuracy of 11 strips, the mean and standard deviation of elevation difference of tie-points between 11 strips obtained using two processing modes are compared, and the elevation difference distribution of all tie-points (80 in total) in 11 strips is calculated. The results are shown in Figure 20.

It can be seen in Figure 20b,d that the elevation differences of the tie-points between strips of different sorties obtained under the two processing modes are normally distributed, and most of the elevation differences are within ±0.5 m. At the same time, by comparing the two processing modes, Figure 20a,c shows that the mean value and variance of elevation difference of tie-points between strips are small, which meets the requirements of high-precision geographic mapping.

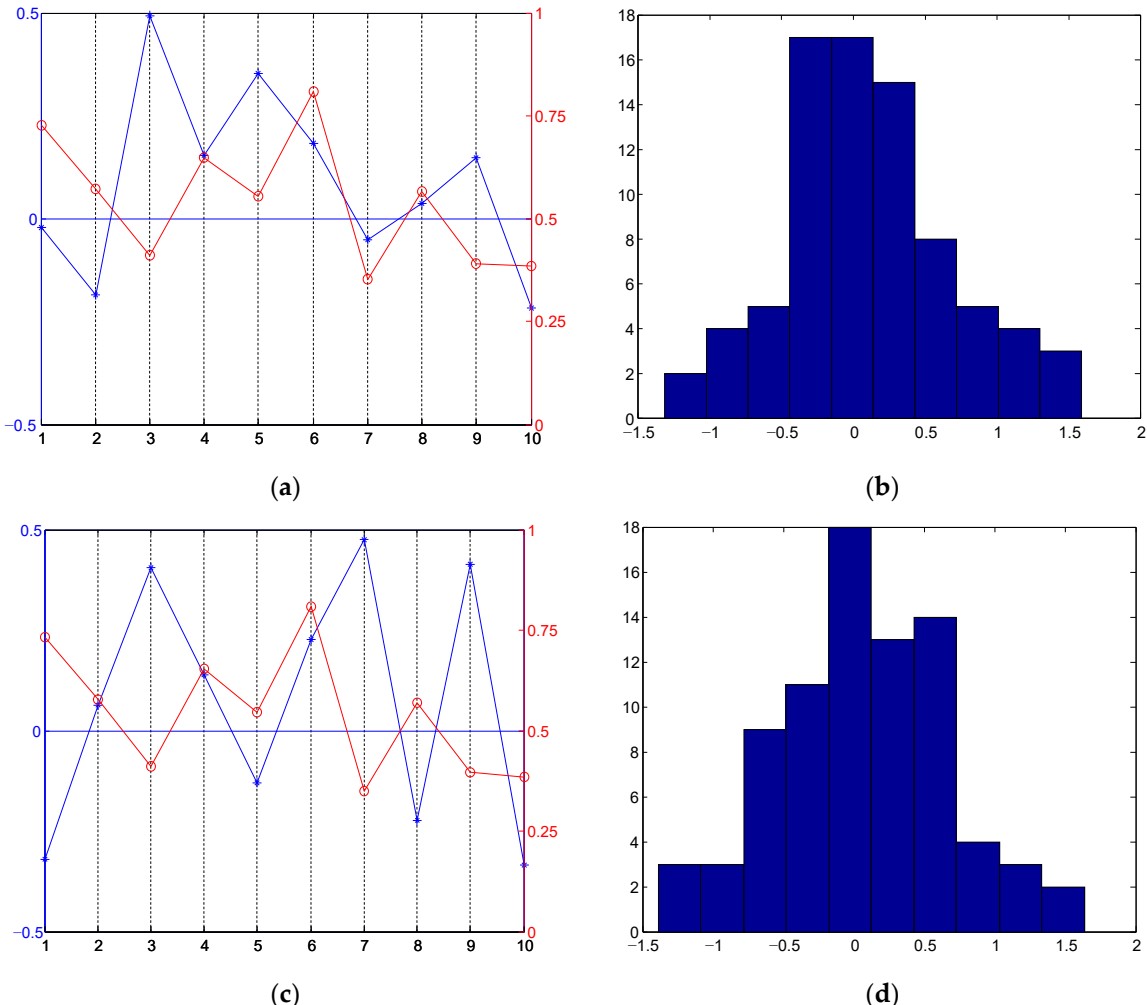

**Figure 20.** Mean value, standard deviation and distribution of elevation difference of tie-points between strips, where (**a**,**b**) are the results based on tie-points, (**c**,**d**) are the results based on marking points. Additionally, the abscissa in the (**a**,**c**) is the number of data stacks between strips, and the left and right ordinate in the (**a**,**c**) are mean elevation difference (blue) and standard deviation (red) of tie-points in data stacks between strips whose unit is m, respectively. Additionally, the abscissa in the (**b**,**d**) is elevation difference (in meters) of data stacks between strips, and the ordinate in the (**b**,**d**) is the number of tie-points corresponding to elevation difference.

## 4. Discussion

### 4.1. Comparsion and Analysis of Elevation Inversion

Section 3 mainly shows the extraction results of tie-points, the interferometric calibration parameters and the elevation inversion results of tie-points, but does not analyze the elevation of the complete data block. In this paper, two adjacent strips, namely strip 5 and strip 6 in Section 3.2.2., are selected, and the elevation information of the whole data block is inversed using the calibration parameters obtained from two modes based on the tie-points and the marking points to verify the effectiveness of the algorithm. The elevation inversion results of the data block in strip 5 are shown in Figure 21a,b, where (a) is DSM inversed based on the tie-points, and (b) is based on marking points. In addition, the difference in two DSM products produced based on the tie-points and marking points and the statistical analysis results are shown in Figure 21c,d. It can be seen from the statistical results that the elevation difference of the DSM products produced by the two modes is normally distributed, with the mean value of −0.03 m, the standard deviation of 0.23 m and most of the elevation differences are within ±1 m. This shows that elevation inversion

based on the tie-points and marking points can obtain high-precision DSM products, and both have high-precision topographic mapping capabilities.

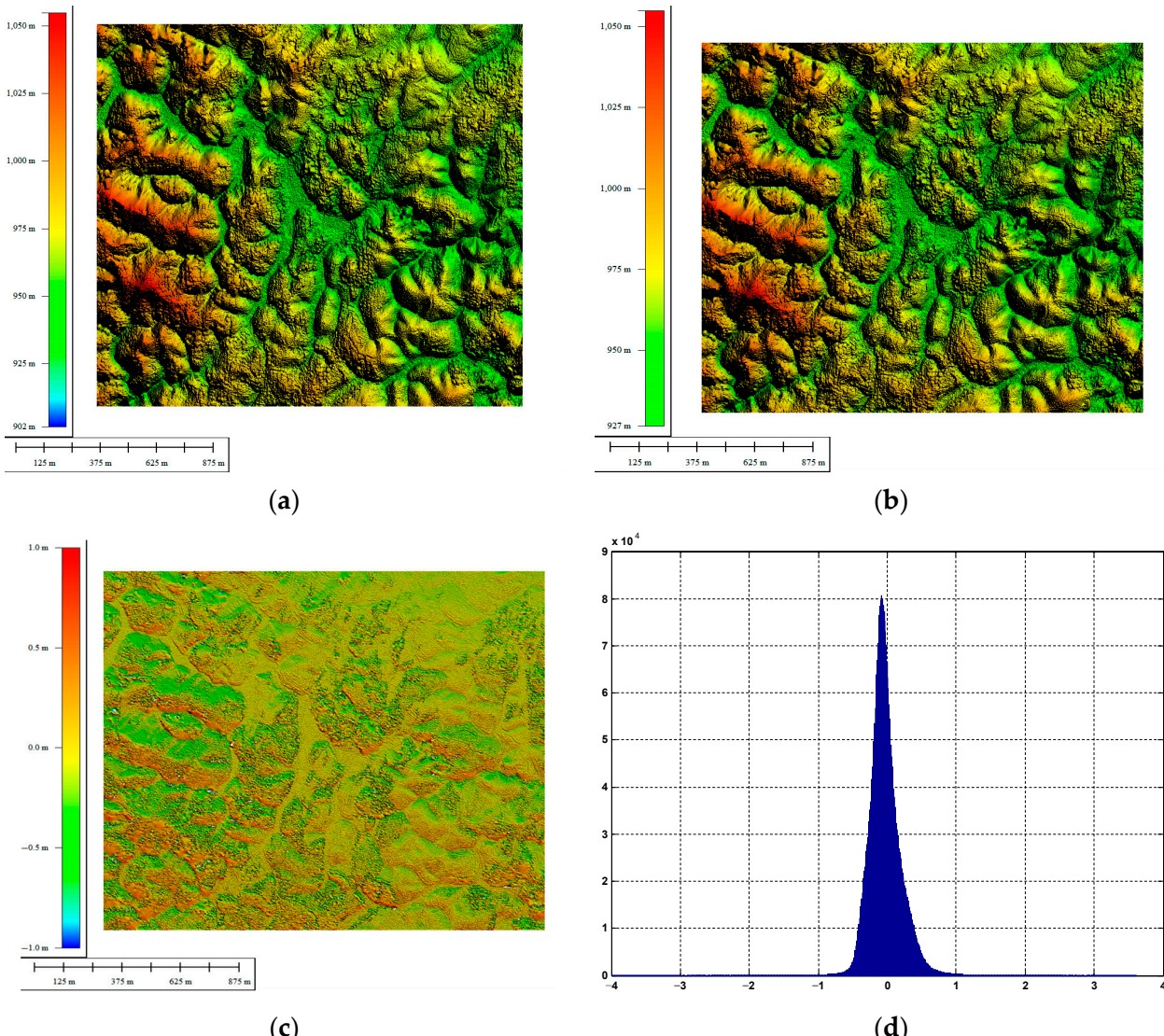

(**a**)

(**b**)

(**c**)

(**d**)

**Figure 21.** DSM products of strip 5 obtained through elevation inversion, where (**a**) is based on the tie-points, (**b**) is based on marking points and (**c**,**d**) are the difference and statistical results of two DSMs. Additionally, the abscissa in (**d**) is elevation difference, whose unit is m, and the ordinate in the (**d**) is the number of pixel points corresponding to relative elevation difference.

Strips 5 and strip 6 are adjacent strips with hilly and mountainous terrain, and the DSM products of strip 6 based on the tie-points' and marking points' interferometric calibration parameters are shown in Figure 22a,b. In order to explore the elevation accuracy between strips, this subsection analyzes the difference in DSM in the overlapping area of strip 5 and strip 6, and obtains the difference images and statistical analysis results as shown in Figure 22c–f, where (c) and (d) are the difference and statistics of DSM products based on the tie-points, and (e) and (f) are based on marking points.

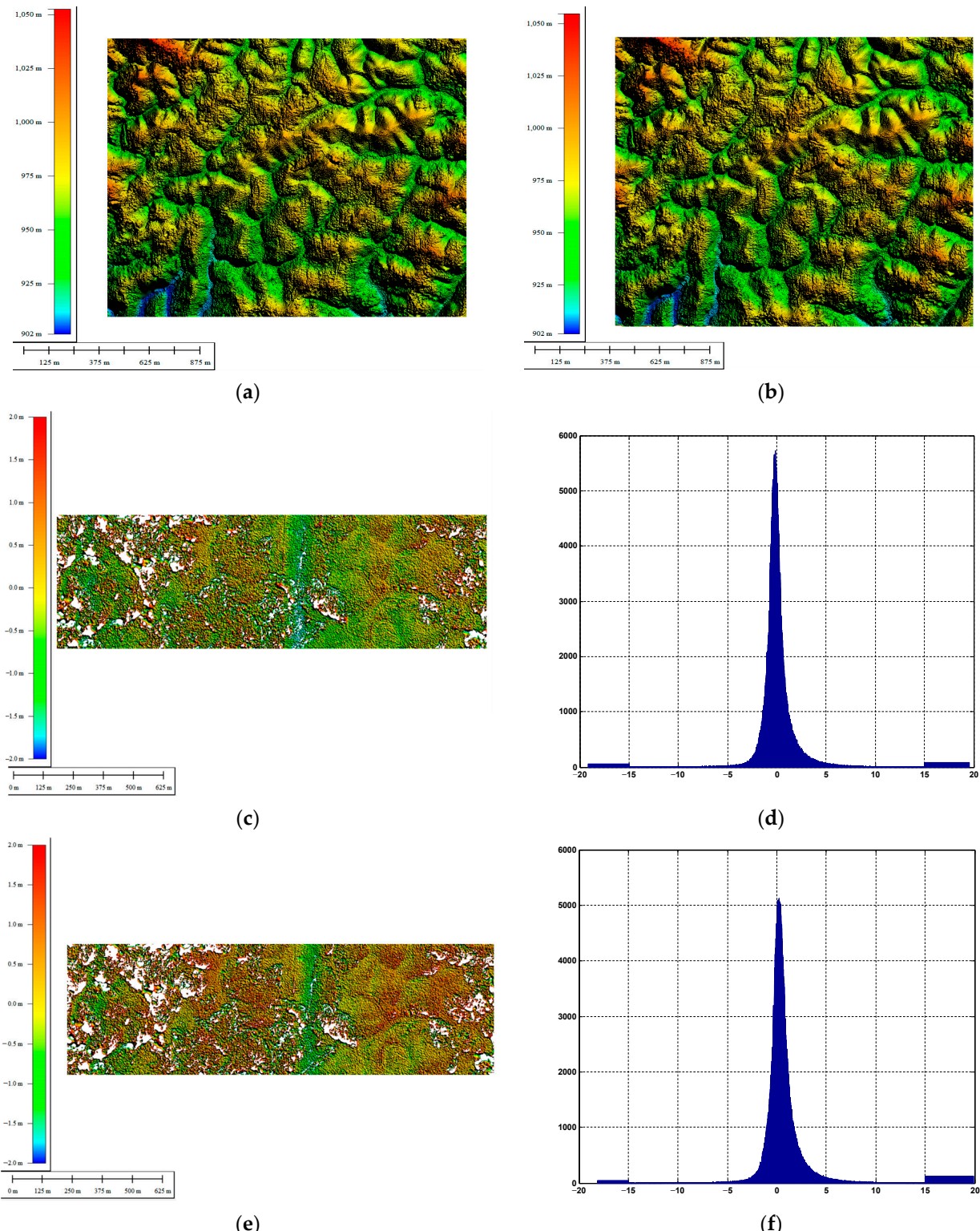

**Figure 22.** DSM products of strip 5 obtained by elevation inversion, where (**a**) is based on the tie-points, (**b**) is based on marking points, (**c**,**d**) are the difference and statistical results of two DSMs based on the tie-points, and (**e**,**f**) are the difference and statistical results of two DSMs based on the marking points. Additionally, the abscissa in (**d**,**f**) is elevation difference, whose unit is m, and the ordinate in (**d**,**f**) is the number of pixel points corresponding to relative elevation difference.

Figure 22d shows that the difference in DSM in the overlapping area between strips based on the tie-points is normally distributed, with a mean value of 0.05 m and standard deviation of 1.48 m. The proportion of the elevation difference in the overlapping area within ±2 m is 90.1% (the area with an error beyond ±10 m is the shadow, which is not considered). Figure 22f shows that the DSM difference in the overlapping area between strips based on marking points also presents a normal distribution, with a mean value of 0.44 mm and standard deviation of 1.55 m. The proportion of the elevation difference in the overlapping area within ±2 m is 87.5%. From the comparison of the two results, the algorithm based on the tie-points can produce high-precision DSM products. The mean value of elevation difference between strips is close to 0, and the strips are excessively flat without elevation jump. In addition, compared with the marking points' calibration algorithm, the DSM product based on the tie-points calibration has smaller variance of elevation difference between strips, indicating that the elevation accuracy is more stable, which further verifies the effectiveness and feasibility of airborne millimeter-wave InSAR topographic mapping technology based on the tie-points.

### 4.2. Special Issues in Elevation Inversion

When using the automatically extracted tie-points for interferometric calibration, if the phase error occurs in an area due to the data itself (for example, the areas of mountains, cities, shadows, etc. account for a large proportion of the data block), the correct elevation transfer will be affected, resulting in errors in the elevation inversion of subsequent data blocks. Typical areas such as water bodies, urban areas and shadows are low coherence areas which impair the effectiveness of normal phase unwrapping. This subsection analyzes the problems in elevation inversion of these three typical areas, and proposes relevant processing measures.

#### 4.2.1. Elevation Inversion of Water Area

The number of pixels in the range direction and azimuth direction of the data block shown in Figure 23a are 8704 and 9702, respectively. It can be clearly seen from the SAR image that the data block contains a large number of water areas which play a role in agricultural remote sensing and environmental monitoring [36,37]. The phase obtained through the traditional phase unwrapping method has phase discontinuity, as shown in Figure 23b, which leads to errors in the elevation transfer of tie-points [38]. The solution to this type of problem is to remove the phase of rough terrain from the filtered phase. Since the terrain around the water body fluctuates little, the phase does not appear in the $2\pi$ wrapping after removing the rough terrain phase, as shown in Figure 23c. Therefore, the phase unwrapping operation is not required, and the phase result after unwrapping can be obtained by directly adding the rough terrain phase, as shown in Figure 23d. It can be found that the normal phase unwrapping is achieved in the land area, and the phase is consistent, so that the tie-points can carry out normal elevation transfer.

#### 4.2.2. Elevation Inversion of Urban Area

The data shown in Figure 24a is a typical urban SAR image, with 8704 and 8352 pixels in range and azimuth direction, respectively. The SAR amplitude image shows that some high buildings in urban areas have amplitude oversaturation, which leads to an obvious side lobe effect. In addition, from the coherence coefficient diagram in Figure 24b, it can be found that there are a large number of low-coherence areas in urban areas due to roads, shadows, multipath scattering and other reasons. The traditional phase unwrapping has obvious errors, as shown in Figure 24c, which not only leads to errors in the elevation inversion of the data block itself, but also leads to errors in the elevation transfer of tie-points, resulting in errors in the elevation inversion of adjacent data blocks.

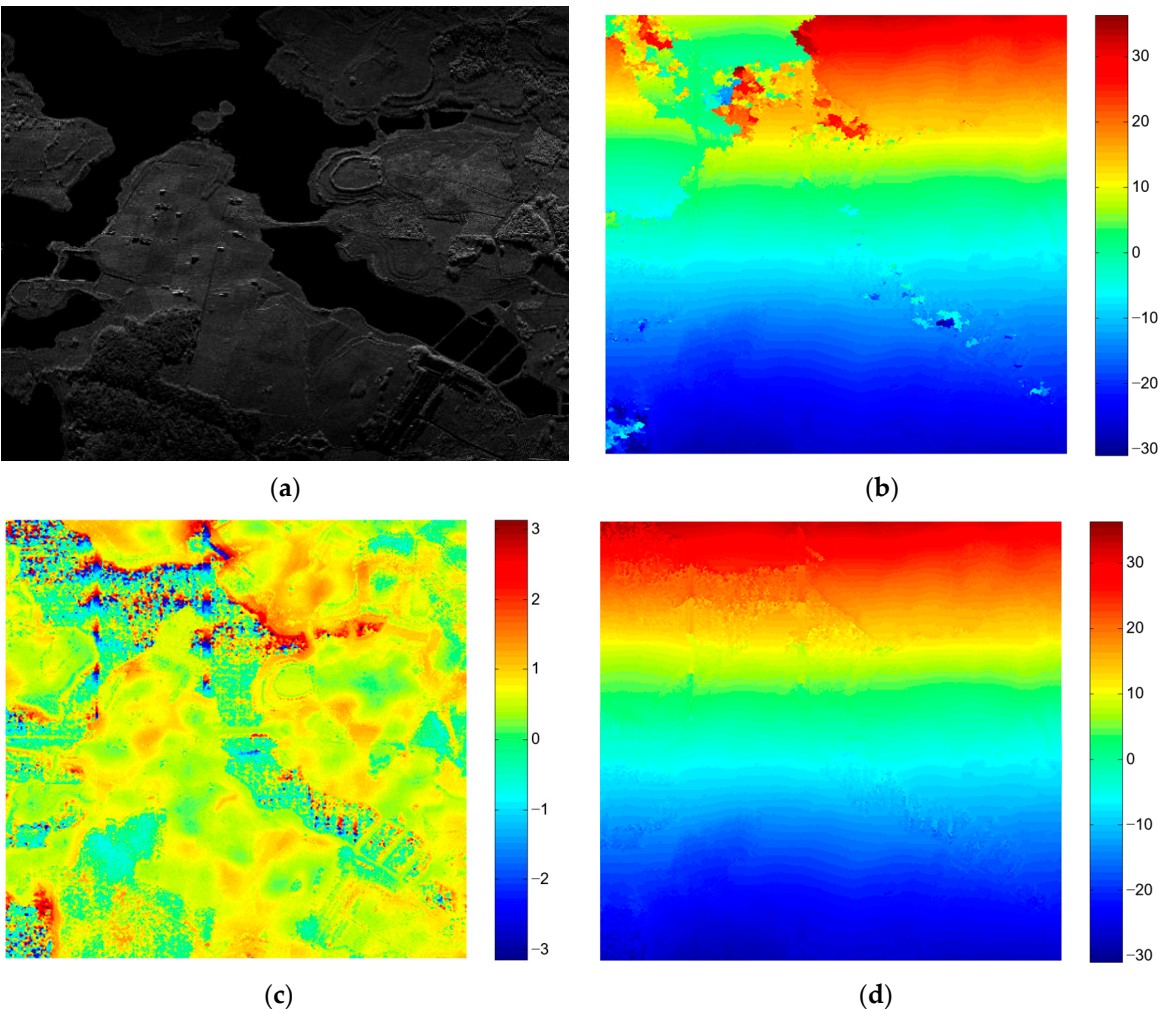

**Figure 23.** Phase processing of data block containing water area, where (**a**) is the amplitude image, (**b**) is the result of conventional phase unwrapping, (**c**) is the result of removing the rough terrain (SRTM) phase, and (**d**) is the correct phase unwrapping result.

In view of the problem that elevation inversion in urban areas is prone to errors, the method of combining long and short baselines is adopted to overcome this. First, the short baseline data are used to generate the rough terrain, then the phase of the rough terrain is subtracted from the phase of the long baseline data, and then the correct phase is obtained using phase unwrapping. The phase unwrapping result of the long baseline obtained by this method is shown in Figure 24d. It can be seen from the figure that the phase in the left and right of the image is consistent and effective for the subsequent elevation inversion. Further processing details and results are available in [39].

4.2.3. Elevation Inversion of Mountain Area

The data block shown in Figure 25a is a typical SAR image of mountain area, and the number of pixels in the range and azimuth directions are, respectively, 11,008 and 19,040. The mountain area is characterized by gullies, large topographic relief and large areas of layover and shadow areas, resulting in a large number of low coherence areas. which makes obvious errors in the traditional phase unwrapping [40]. The coherence coefficient and phase unwrapping results are shown in (b) and (c) of Figure 25, respectively. In addition, due to the existence of shadows, there are fewer real objects in the overlapping area of adjacent data blocks in mountain areas, which makes it difficult to automatically extract tie-points.

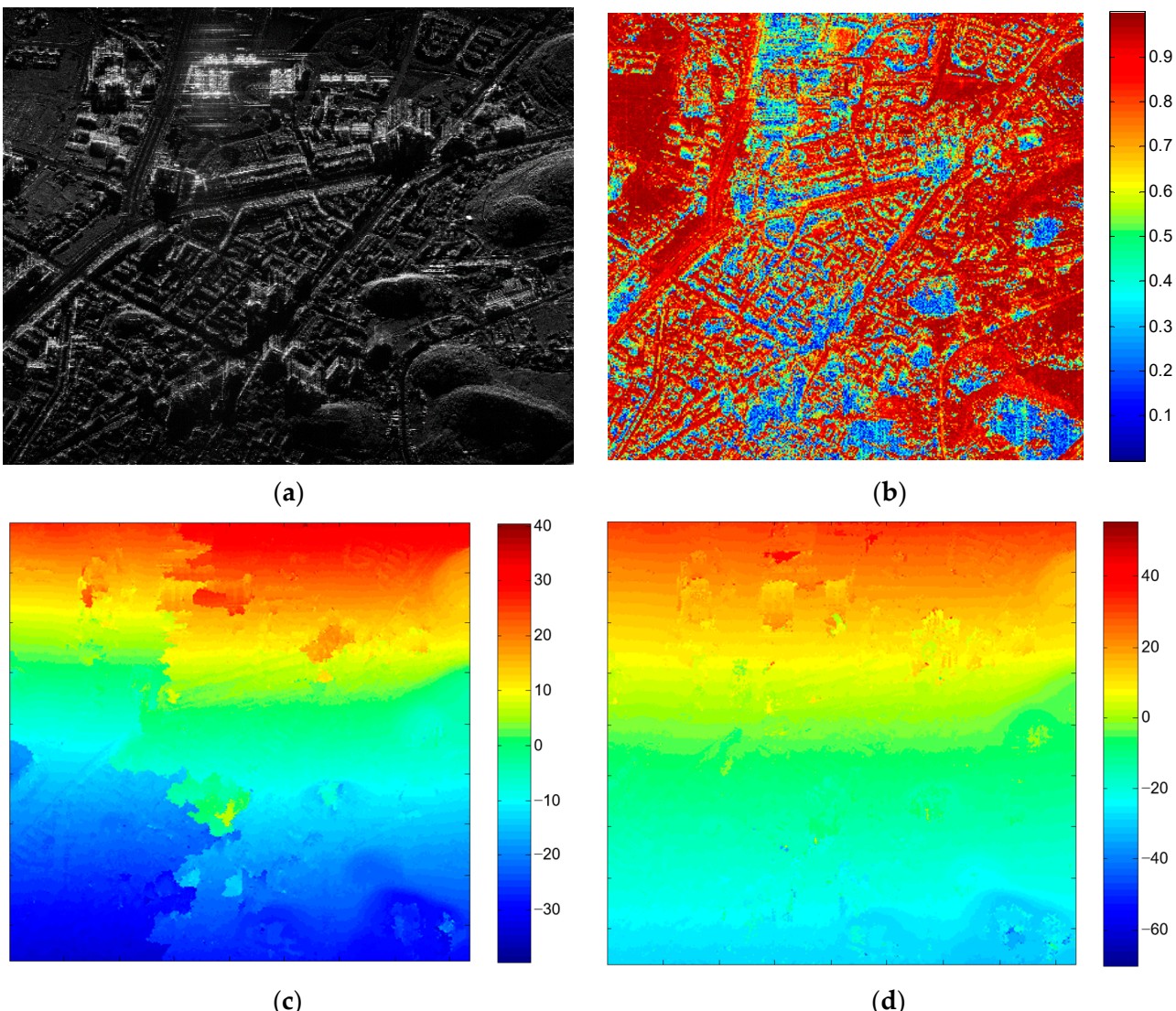

**Figure 24.** Phase processing and elevation inversion of data block containing urban area, where (**a**) is the amplitude image, (**b**) is coherence coefficient diagram, (**c**) is the result of conventional phase unwrapping and (**d**) is the correct phase unwrapping result.

In view of the problems that elevation inversion of mountain areas is prone to errors and tie-points are difficult to extract, as in urban areas, the method of combining long and short baselines is adopted for processing. The phase unwrapping results using short baseline and long baseline data are shown in Figure 25d and e, respectively, and the elevation inversion results are shown in Figure 25f. From the elevation inversion results, the terrain fluctuation of the data block is more than 400 m, and the elevation inversion of the effective surface features in the mountain area is correct, which shows that the data processing method based on rough terrain and long-short baselines can realize the elevation inversion of complex mountain areas. In addition, due to the complex terrain in mountain areas, a large number of layover and shadow areas appear in SAR data. Therefore, it is necessary to supplement invalid data by means of multi angle flight, such as pair flight and vertical flight. These topics are not the focus of this study and will be discussed in depth in the future.

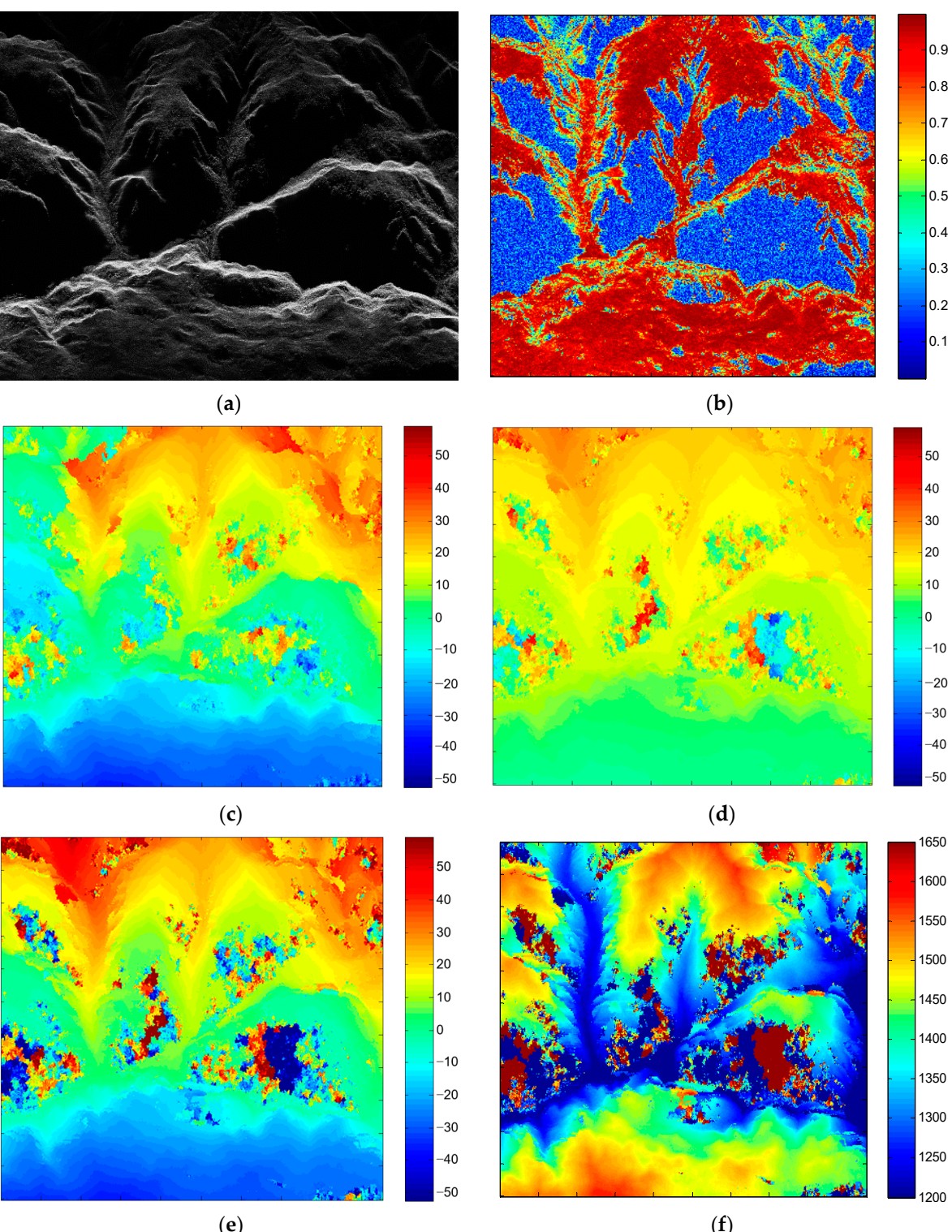

**Figure 25.** Phase processing and elevation inversion of data block containing shadow area, where (**a**) is the amplitude image, (**b**) is coherence coefficient diagram, (**c**) is the result of conventional phase unwrapping, (**d**) is the phase unwrapping result of short baseline, (**e**) is the phase unwrapping result of long baseline and (**f**) is the elevation inversion result.

## 5. Conclusions

In this paper, firstly, based on the principle of SAR-SIFT, the automatic extraction of tie-points in the overlapping area between adjacent images is realized, then airborne InSAR elevation inversion is realized through the interferometric calibration based on the

tie-points. The processing results of the measured data show that the method in this paper can effectively solve the problem of large-area high-precision topographic mapping, and has good engineering application value.

In addition, based on the processing results of the measured data, this paper puts forward relevant suggestions on airborne millimeter-wave InSAR flight and data processing:

(1) During flight route design, ensure that one sortie covers as many continuous strips as possible, and avoid consecutively adjacent strips being distributed in multiple sorties, which results in increased errors in elevation transfer of adjacent strips, leading to errors in elevation inversion.

(2) If corner reflectors need to be deployed on the ground during flight, they can be deployed across two sorties to reduce the workload.

(3) The initial phase and other parameters are consistent when processing different strips of the same sortie, so the SRTM reference data can be used to directly calculate the phase winding N value of each data block without extracting the tie-points when the height of ambiguity in the airborne InSAR system is larger than the height error between SRTM and real terrain elevation. If there is no SRTM data, the short baseline data can be used to quickly calculate the coarse DEM to replace the SRTM data.

(4) When selecting the tie-points between strips of different sorties, the positions of the tie-points should be accurate, the number should be about 20, and the slant range error should be less than one ground resolution, so as to ensure the accurate transmission of error parameters of strips between different sorties.

In the future, we will continue to carry out automatic identification of abnormal data blocks such as water bodies, urban areas and mountain shadows, automatic extraction and distribution optimization of tie-points between strips, and research on modeling and processing of joint block adjustment so as to improve the robustness of the algorithm in this paper. We aim to further realize fully automatic less-/no-control high-precision mapping processing of airborne InSAR, and improve the overall accuracy of airborne InSAR large-scale mapping.

**Author Contributions:** Conceptualization, S.L. and F.X.; methodology, B.Z.; validation, B.Z. and L.W. (Liuliu Wang); formal analysis, L.F.; investigation, L.W. (Lideng Wei); data curation, L.W. (Liuliu Wang); writing—original draft preparation, B.Z. and F.X.; writing—review and editing, F.X.; visualization, L.W. (Liuliu Wang); supervision, S.L.; project administration, L.F.; funding acquisition, L.W. (Lideng Wei). All authors have read and agreed to the published version of the manuscript.

**Funding:** This research was funded by the Key R&D Program Projects in Hainan Province (No. ZDYF2019008), the State Key Laboratory of Rail Transit Engineering Information (SKLK22-08) and Key Projects of the Ministry of Science and Technology of China.

**Data Availability Statement:** Not applicable.

**Acknowledgments:** The authors would like to thank the editors and the anonymous reviewers for their valuable suggestions.

**Conflicts of Interest:** The authors declare no conflict of interest.

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
