# Peer review of "Airborne Millimeter-Wave InSAR Terrain Mapping Experiments Based on Automatic Extraction and Interferometric Calibration of Tie-Points"

_remotesensing, doi:10.3390/rs15030572_

Round 1
Reviewer 1 Report
Comments can be found in the attachment.

Author Response
Dear Reviewer,
We really appreciate the reviewer’s help very much not only for the suggestions, but also for the patience and kindness to point out some existing problems. And we are very sorry for the various problems in the illustration of our proposal in the paper, and thank you very much for your careful reading. We have made these corrections and added more explanations in revised manuscript. The positive comment indeed encourages us for a further study on the InSAR Terrain Mapping of airborne millimeter-wave InSAR.
The attachment is our response to the reviewer’s comments.
Yours sincerely.

Reviewer 2 Report
The paper proposes an automatic extraction of tie-points and interferometric calibration technology. It uses the automatic extraction algorithm of tie-points based on SAR+SIFT+RANSAC to obtain the tie-points of adjacent images. The measured data is used to verify the technology, and compares it with the areas with control points and marking points.
The article is very well written, with significant theoretical development and significant simulation results supporting the proposed idea.
However, the novelty compared to existing work is not clearly demonstrated
Author Response
Dear Reviewer,
We really appreciate the reviewer’s help very much not only for the suggestions, but also for the patience and kindness to point out some existing problems. We have made these corrections and added more explanations in the revised manuscript. The positive comment indeed encourages us for a further study on the InSAR Terrain Mapping of airborne millimeter-wave InSAR.
The attachment is our response to reviewer’s comments.
Yours sincerely.

Reviewer 3 Report
1-this paper proposes an automatic extraction of tie-points and interferometric calibration technology. And this paper discusses the difficulties in the treatment of typical areas, such as water areas, urban areas and mountain areas, and gives reasonable solutions,
2 The topic is original and relevant in the Calibration of Tie-Points.
3. When compared with other published papers ;The processing results of the measured data show that the method in this paper can effectively solve the problem of large area high-precision topographic mapping.
4. when consider regarding the methodology?
4.1 Where is B at figüre 1 Please Show it on figüre 1
4.2 İn line 101 :
2.1. Fundamentals of InSAR and Problem Despriction
It should be like this
2.1 Fundamentals of InSAR and Problem Description
4.3 What is DSM please explain it at methodology section
4.4 Please explain SAR processing procedures at methodology section
5. presented conclusions are consistent with the evidence and arguments
6. I would like to recommend some appropriate references
Duysak, H. & Yiğit, E. (2022). Investigation of the performance of different wavelet-based fusions of SAR and optical images using Sentinel-1 and Sentinel-2 datasets . International Journal of Engineering and Geosciences , 7 (1) , 81-90 . DOI: 10.26833/ijeg.882589
Yağmur, N. , Tanık, A. , Tuzcu, A. , Musaoğlu, N. , Erten, E. & Bilgilioglu, B. (2020). Opportunities provided by remote sensing data for watershed management: example of Konya Closed Basin . International Journal of Engineering and Geosciences , 5 (3) , 120-129 . DOI: 10.26833/ijeg.638669
Polat, A. B., Balik Sanli, F., & Akcay, O. (2022). Analyzing rice farming between sowing and harvest time with Sentinel-1 SAR data. Advanced Remote Sensing, 2(1), 34–39. Retrieved from https://publish.mersin.edu.tr/index.php/arsej/article/view/248
Author Response
Dear Reviewer,
We really appreciate the reviewer’s help very much not only for the suggestions, but also for the patience and kindness to point out some existing problems. We have made these corrections and added more explanations in revised manuscript. The positive comment indeed encourages us for a further study on the InSAR Terrain Mapping of airborne millimeter-wave InSAR.
The attachment is our response to reviewer’s comments.
Yours sincerely.

Round 2
Reviewer 1 Report
I would like to appreciate the authors' patience with the clarification and the modifications in the reply. I hope other readers share the same delightful reading experience with the current manuscript. The present manuscript is recommended for acceptance.